# Structure and biochemistry-guided engineering of an all-RNA system for DNA insertion with R2 retrotransposons

KeHuan K. Edmonds [1,2,3,4,5,6,7], Max E. Wilkinson[1,2,3,4,5,6,7], Daniel Strebinger [1,2,3,4,5,6,7], Hongyu Chen [4,8], Blake Lash [1,2,3,4,5,6,7], Clarissa C. Schaefer [1,2,3,4,5,6,7], Shiyou Zhu[1,2,3,4,5,6,7], Dangliang Liu [4,8], Shai Zilberzwige-Tal[1,2,3,4,5,6,7], Alim Ladha[1,2,3,4,5,6,7], Michelle L. Walsh[1,2,3,4,5,6,7], Chris J. Frangieh[1,2,3,4,5,6,7], Nicholas A. Vaz Reay[1,2,3,4,5,6,7], Rhiannon K. Macrae [1,2,3,4,5,6,7], Xiao Wang[4,8,9] & Feng Zhang [1,2,3,4,5,6,7] ✉

R2 elements, a class of non-long terminal repeat (non-LTR) retrotransposons, have the potential to be harnessed for transgene insertion. However, efforts to achieve this are limited by our understanding of the retrotransposon mechanisms. Here, we structurally and biochemically characterize R2 from *Taeniopygia guttata* (R2Tg). We show that R2Tg cleaves both strands of its ribosomal DNA target and binds a pseudoknotted RNA element within the R2 3′ UTR to initiate target-primed reverse transcription. Guided by these insights, we engineer and characterize an all-RNA system for transgene insertion. We substantially reduce the system's size and insertion scars by eliminating unnecessary R2 sequences on the donor. We further improve the integration efficiency by chemically modifying the 5′ end of the donor RNA and optimizing delivery, creating a compact system that achieves over 80% integration efficiency in several human cell lines. This work expands the genome engineering toolbox and provides mechanistic insights that will facilitate future development of R2-mediated gene insertion tools.

The ability to perform targeted kilobase gene insertions in the human genome would open up new avenues in medicine and research, enabling installation of transgenes to correct genetic diseases or modulate cell behaviors. A number of approaches to insert genes have been described, many of which capitalize on natural machinery, such as knock-in systems based on CRISPR, transposons, recombinases, and retroviruses[1–6]. However, most systems randomly integrate the transgene cargo and/or require exogenous DNA, thereby elevating innate immune responses[7–9].

An all-RNA targeted gene insertion system can overcome these limitations. To achieve this, we investigate another natural machinery, the non-long terminal repeat (non-LTR) R2 retrotransposon. R2 elements propagate via a copy-and-paste mechanism targeting the 28S ribosomal RNA (rRNA) gene[10,11], which exist in more than 100 copies in the average human genome[12] and can serve as a safe harbor for transgene expression[13–15]. To insert a new copy of itself, the R2 protein recruits the R2 RNA and nicks the bottom strand of the 28S DNA. Using the nicked DNA as a primer and the R2 RNA as a template, it reverse

[1]Howard Hughes Medical Institute, Cambridge, MA, USA. [2]Yang Tan Collective, Cambridge, MA, USA. [3]McGovern Institute for Brain Research at MIT, Cambridge, MA, USA. [4]Broad Institute of MIT and Harvard, Cambridge, MA, USA. [5]Department of Brain and Cognitive Science, Massachusetts Institute of Technology, Cambridge, MA, USA. [6]Department of Biological Engineering, Massachusetts Institute of Technology, Cambridge, MA, USA. [7]Department of Stem Cell and Regenerative Biology, Harvard University, Cambridge, MA, USA. [8]Department of Chemistry, Massachusetts Institute of Technology, Cambridge, MA 02139, USA. [9]Stanley Center for Psychiatric Research, Broad Institute of MIT, Cambridge, MA 02142, USA. ✉e-mail: zhang@broadinstitute.org

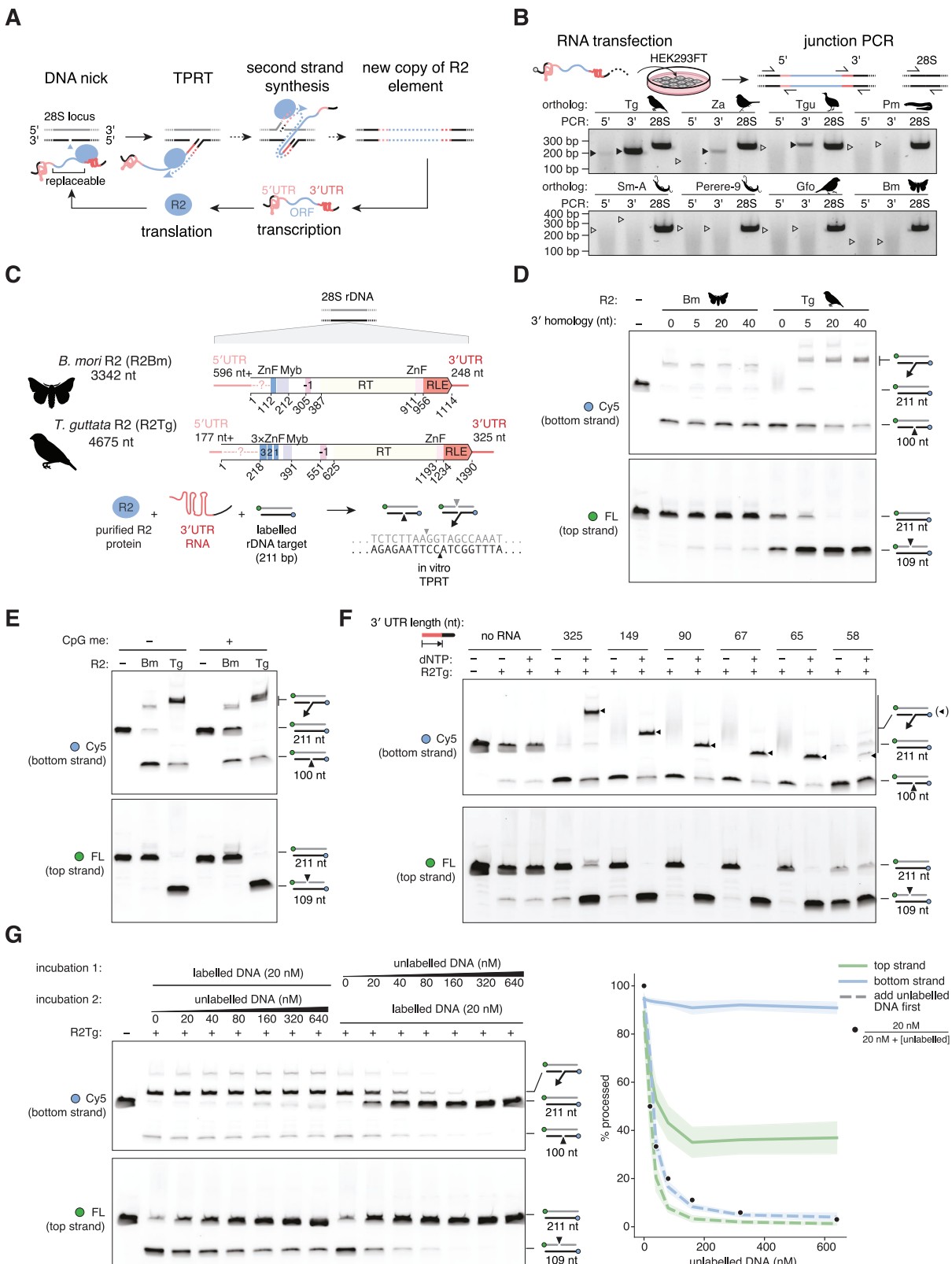

transcribes a new copy of the element – a process known as target-primed reverse transcription (TPRT). The protein also nicks the top strand of the target, likely through a reverse transcription-dependent mechanism[16]. The subsequent steps remain poorly understood: the nicked top strand presumably serves as a primer for second strand synthesis by R2 reverse transcriptase's DNA-dependent DNA polymerase activity and/or a host polymerase, using the newly synthesized

first strand DNA as a template[17,18]. The DNA break and integration is then likely resolved by the endogenous repair machinery of the cell[10] (Fig. 1A). Recent structural and biochemical studies suggest that only the untranslated regions (UTRs) of the R2 RNA are required for the retrotransposon-mediated integration[16,19]. Thus, it is possible to create an all-RNA knock-in system by replacing the open reading frame (ORF) in the R2 RNA with a transgene of interest (Fig. 1A). Indeed, several

**Fig. 1 | Biochemical characterization of R2Tg. A** Schematic of natural R2 system propagation. The non-LTR retrotransposon binds to the UTR of its RNA species, nicks the 28S DNA, and synthesizes a new copy of the R2 element through TPRT. **B** Comparison between different R2 orthologs in human cells (repeated two additional times with similar results). Tg, *Taeniopygia guttata* (R2-A); Za, *Zonotrichia albicollis* (R2-A); Tgu, *Tinamus guttatus* (R2-A); Pm, *Petromyzon marinus* (unknown clade, likely R2-A); Sm-A and Perere-9, *Schistosoma mansoni* (R2-C); Gfo, *Geospiza fortis* (R2-A); and Bm, *Bombyx mori* (R2-D). Arrows indicate the expected PCR product sizes. "28S" products were diluted 15-fold relative to the others prior to gel electrophoresis. **C** Domains of the R2Bm and R2Tg retrotransposons. The 5′ UTR boundaries are not known precisely due to the uncertain start codon position. The ORF amino acids are numbered from the earliest plausible start codon. ZnF, zinc finger; RT, reverse transcriptase; RLE, restriction-like endonuclease. **D–G** Denaturing gels of in vitro TPRT reactions on a dual-labeled 211-bp 28S DNA target (repeated at least once with similar results). The same gel is visualized by Cy5 fluorescence (bottom strand) and fluorescein fluorescence (top strand). FL, fluorescein. Slower migrating bands above the TPRT product likely correspond to template jumping after TPRT. Smeared bands are likely due to residual denatured R2Tg protein. **E**, with and without prior in vitro CpG methylation of the DNA target using M.SssI CpG methyltransferase. **F**, with truncations from the 5′ side of the R2Tg 3′ UTR, which maintains 40 nt of 3′ homology. **G**, with two sequential incubations with unlabeled or dual-labeled DNA substrate prior to dNTP addition. This assay used the 149 nt R2Tg 3′ UTR with 40 nt of 3′ homology. Right: quantification of gel bands (*n* = 3, data presented as mean±standard error). % processed for the top strand is calculated as intensity cleaved divided by total lane intensity. % processed for the bottom strand is calculated as intensity of the cleaved band plus intensity of the TPRT band, divided by the total lane intensity. Source data are provided as a Source Data file.

## Results

### Native R2Tg is active in human cells

groups have reported some success with this in zebrafish and mammalian cells[20–22]. However, the limited insertion efficiency to date and the lack of in vivo data underscore the need for deeper mechanistic understanding of both the natural and engineered R2 systems.

Here, we show that the R2 element from the zebra finch *Taeniopygia guttata* (R2Tg) is active in human cells and can be engineered for efficient transgene insertion. We biochemically characterize the retrotransposon R2Tg and report its cryo-electron microscopy (cryo-EM) structure as it initiates TPRT, highlighting key differences from the more extensively studied R2 from the silk moth *Bombyx mori* (R2Bm). Leveraging these insights, we adapt R2Tg to integrate non-R2 cargo RNA in mammalian cells and systematically identify and resolve efficiency-limiting factors. The engineered system provides a compact and chemically modified all-RNA system that performs robustly in multiple mammalian cell types.

## Results

### Native R2Tg is active in human cells

We first screened R2 systems from different species to identify an ortholog with high basal activity in human cells to serve as the basis for our system. We selected orthologs from the R2-A, -C, and -D clades and evaluated their ability to integrate their respective R2 RNAs in the 28S locus of HEK293FT cells (Supplementary Fig. 1A)[11,23,24]. We synthesized the R2 RNAs containing ORFs flanked by their endogenous UTRs through in vitro transcription (IVT). To reduce signal from homology-directed repair (HDR), we elected to use short homologies to the 28S locus: 36-nt homology on the 5′ end, which is sufficient to form the hepatitis delta virus (HDV)-like ribozymes found on many known R2 orthologs[25], and 5-nt homology on the 3′ ends of the transcript, based on the number of template-primer base pairs resolved in the structure of R2Bm and group II introns[19,26,27]. To facilitate RNA translation and stability in mammalian cells, we synthesized the RNA with standard modifications: a 5′ cap analog[28], N[1]-methylpseudouridine-5′-triphosphate (m1ψ), and a poly(A) tail[29], which incidentally increases the 3′ homology to 8 nt.

We transfected the RNA into HEK293FT cells and harvested genomic DNA (gDNA) one day later to assess integration via junction PCR (Fig. 1B). R2-A orthologs from *T. guttata*, *Zonotrichia albicollis* (white-throated sparrow), and *Tinamus guttatus* (white-throated tinamou) showed detectable activity, with R2Tg showing the highest level, in agreement with other recent reports[21,22]. Based on these results, we proceeded to characterize the most active candidate, R2Tg.

### Biochemical characterization of R2Tg TPRT activity

Most of the insights into the biochemical mechanisms of R2 elements stem from work on R2Bm. Previous studies established that the R2Bm ORF uses N-terminal Myb and zinc-finger DNA binding domains to recognize a 13-bp sequence motif upstream of the ribosomal DNA (rDNA) target site and the RT − 1 domain to bind a two-hairpin element within the R2Bm 3′ UTR RNA[16,19,30,31]. The target site is unwound and the

bottom strand is nicked by the restriction endonuclease-like (RLE) domain, and the nicked bottom strand acts as a primer for reverse transcription of the full R2Bm RNA. A structured element near the 5′ end of the R2Bm RNA can promote cleavage of the rDNA top strand by the RLE domain[16,32]. Top strand cleavage is presumably important to create a primer for second strand synthesis and completion of retrotransposition, but the mechanistic details remain poorly understood.

The R2Tg element is a member of the R2-A clade common in birds, with several differences compared to R2Bm, which is in the R2-D clade[33]. R2-A ORFs encode three N-terminal zinc finger domains, whereas R2-D ORFs encode one (Fig. 1C). Moreover, the 5′ and 3′ UTR sequences have little apparent similarity to the conserved core elements of the R2Bm UTRs, with the exception of an HDV-like ribozyme at the 5′ end[25]. Therefore, the details of R2Tg interactions with self RNA and target DNA may be different from the well characterized interactions of R2Bm. To investigate this point, we expressed the R2Tg ORF in *Saccharomyces cerevisiae* as a GFP fusion, purified it using a GFP nanobody column, and compared its properties to the R2Bm ORF expressed and purified from *E. coli* as described previously[19].

We first assayed for TPRT activity by mixing R2Bm or R2Tg with their corresponding 3′ UTR RNAs (248 nt for R2Bm, 325 nt for R2Tg) and a 211-bp labeled DNA substrate containing the 28S target sequence (Fig. 1D). In the absence of 3′ homology, R2Bm showed efficient bottom strand cleavage and TPRT activity, while the top strand was unprocessed, consistent with previous results[16,32]. R2Tg only showed bottom strand cleavage, without detectable TPRT activity. The natural 3′ end of R2 RNAs produced in host cells is not known, but in the absence of a specific nuclease or termination activity it is expected to extend into the 28S ribosomal RNA itself, producing complementarity to the nicked bottom strand (hereafter, 3′ homology). 3′ homology was previously shown to minimally affect R2Bm TPRT activity[26]. However, when we added increasing amounts of 3′ homology to the R2Tg 3′ UTR we found increasing stimulation of TPRT activity. We conclude that in contrast to R2Bm, R2Tg TPRT activity strictly requires 3′ homology.

Compared to *B. mori*, both mammalian and avian genomes have extensive CpG methylation[34–36]. To determine if this may affect the R2 systems' ability to access its genomic target in human cells, we methylated the CpG sites of the 28S 211-bp DNA substrate and performed TPRT assays with R2Bm and R2Tg (Fig. 1E). Bottom strand cleavage by R2Bm was noticeably inhibited by CpG methylation, whereas R2Tg activity was not affected by DNA methylation. Given that different biological processes and cell types can influence mammalian 28S methylation levels[37,38], R2Tg's ability to cleave the target DNA independent of methylation status may allow it to function in more contexts than its *B. mori* counterpart.

Surprisingly, we noticed that R2Tg exhibited efficient top strand cleavage, particularly when supplied with 3′ UTR RNAs with extended 3′ homology, with efficiencies approaching ~97% after 30 min incubation (Fig. 1D). R2Bm under the same conditions can cleave up to 8% of the top strand. Such pronounced top strand cleavage has not been

observed before in an R2 element[19], so we further investigated this phenomenon. To test if a specific region of the 3′ UTR RNA promotes top strand cleavage, we performed a series of truncations from the 5′ end of the 3′ UTR, while maintaining 40 nt of 3′ homology on the 3′ end, and assayed for TPRT activity and top strand cleavage (Fig. 1F). All RNAs were efficiently used for bottom strand TPRT, up until the 3′ UTR length dropped below 65 nt, at which point TPRT activity was much weaker. However, top strand cleavage was maintained for all RNAs. Intriguingly, for RNAs that promoted TPRT, top strand cleavage was dependent on dNTPs. For example, when R2Tg was incubated with target DNA and a full 3′ UTR RNA without dNTPs, the bottom strand was 92% cut, while the top strand was only 16% cut (Fig. 1F). Upon addition of dNTPs, the bottom strand was extended into the TPRT product, and the top strand was 86% cut. When the 3′ UTR length dropped below 65 nt, top strand cleavage became dNTP-independent, i.e. occurring with 80% efficiency irrespective of the presence of dNTPs, while the bottom strand was 90% cleaved but not extended with or without dNTPs.

R2 proteins only have one endonuclease active site, which in the structure of R2Bm initiating TPRT contained single-stranded bottom strand DNA[19]. We hypothesized that for R2Tg, 3′ UTR RNAs would maintain the protein in a conformation competent for bottom strand cleavage, and that reverse transcription during TPRT would unfold the 3′ UTR and trigger exchange of the DNA substrate within the endonuclease active site. RNAs that are not used for TPRT would not lock R2Tg into a bottom-strand cleavage conformation, allowing dNTP-independent top strand cleavage. To test this hypothesis, we performed a competition assay, in which R2Tg was incubated with a fixed concentration of labeled rDNA before or after incubation with a concentration series of unlabeled rDNA, and after both incubations dNTPs were added (Fig. 1G). If the dissociation rate of the bound DNA is faster than the 30 min timescale of the experiment, unlabeled DNA would be expected to exchange with labeled DNA and result in reaction inhibition. We found that cleavage and TPRT of the bottom strand DNA could not be inhibited when unlabeled DNA was added second, whereas initial incubation with unlabeled DNA strongly inhibited labeled bottom strand activity. This is consistent with R2Tg DNA binding locking it in a TPRT-competent conformation, with the bound target DNA unable to be exchanged with other target DNA in the solvent. This was not observed with top strand cleavage: unlabeled DNA was readily able to inhibit top strand cleavage when added second, reducing labeled top strand cleavage efficiency to ~40%. This result could be explained by a conformational change during or after TPRT in which the target DNA becomes more loosely bound and exchangeable with solvent unlabeled DNA.

## Structural basis for TPRT initiation by R2Tg

To understand differences in the DNA and RNA-binding properties of R2Tg and R2Bm, we incubated R2Tg protein with a biotinylated DNA substrate and 149 nt of 3′ UTR RNA + 40 nt of 3′ homology in the presence of dideoxythymidine to stall TPRT after reverse transcription of a single nucleotide. We purified stalled TPRT complexes using streptavidin beads and solved their structure by cryo-EM to 3.2 Å resolution (Fig. 2A, B, Supplementary Figs. 1-3). The R2Tg TPRT complex has a similar architecture to the R2Bm TPRT complex (Fig. 2C): the target DNA is bound upstream of the target site, and is unwound by a C-terminal zinc finger domain to thread the bottom strand into the RLE domain[19]. The nicked bottom strand is annealed to the RNA 3′ homology within the RT domain, and the 3′ UTR is bound by the RT − 1 domain. However, there are notable differences between the structures, outlined below.

The R2Bm 3′ UTR is 248 nt, and in the R2Bm TPRT structure a 40-nt core element was observed, consisting of two double-stranded regions (P1 and P2) and their single-stranded linker (J1/2), with the complementary regions of P1 and P2 separated by 81 and 98 nt each[19].

In the 325-nt R2Tg 3′ UTR, we similarly found two double-stranded regions with a single-stranded linker, but these were contiguous in sequence space, directly forming the 3′ 67 nt of the 3′ UTR, consistent with our TPRT assays on truncated substrates (Fig. 2D). This region is highly conserved in avian R2 elements (Supplementary Fig. 4). The first of these double-stranded regions, equivalent to P1 in R2Bm, forms a pseudoknot with three base triples, resulting in the ordered region of the R2Tg 3′ UTR being larger and more extensive than the R2Bm 3′ UTR. Nonetheless, there are conserved structural features between the R2Tg and R2Bm 3′ UTRs (Fig. 2E). Although it lacks a pseudoknot, the R2Bm 3′ UTR shares two of the base triples with R2Tg, albeit with different pairing modalities. Both 3′ UTRs have a kink in the J1/2 linker created by an A-A cis Watson-Crick/sugar edge base pair. Both 3′ UTRs also have an A-rich single-stranded tract following P2 that leads into the RT active site. These common structural features allow recognition of the 3′ UTR by a similar RT − 1 domain in the R2 protein, despite the little noticeable primary sequence similarity between R2Tg and R2Bm 3′ UTRs.

We previously showed that R2Bm recognizes target DNA via two sequence motifs: the Retrotransposon Upstream Motif (RUM) 22–34 bp upstream of the bottom strand cleavage site, and the Retrotransposon Associated Site of Insertion (RASIN) directly around the cleavage site[19]. The RASIN is recognized by the RLE domain, while the RUM is recognized by two N-terminal DNA binding domains—a Myb and ZnF domain—and by an R2-specific "RT6a" loop in the RT domain. R2Tg differs from R2Bm in having two additional N-terminal ZnF domains (numbered 2 and 3; Fig. 1C). In the structure of the R2Tg TPRT complex, we noticed differences in RUM recognition to R2Bm (Fig. 2F). The Myb domain, which binds the start of the RUM in R2Bm, was not resolved in our cryo-EM density despite being present in our expression construct. The ZnF domain in common with R2Bm (ZnF 1) inserts an alpha helix into the minor groove of the target DNA, similar to R2Bm, but forms hydrogen bonds to different bases. In R2Bm, Arg125 from the start of the helix recognizes bases −23 (G) and −22 (A), while in R2Tg Gln312 and Arg315 from the end of the helix recognize bases −21 (G) and −20 (T) (numbered relative to the bottom strand cleavage site). Similarly, the RT6a loop inserts into the major groove as in R2Bm, but recognizes different bases. In R2Bm, His673 recognizes bases −27 (G) and −26 (C), and Lys675 recognizes base −24 (G). In R2Tg, Lys922 recognizes bases −25 (G) and −24 (G), and Asp923 recognizes base −23 (C). Therefore, although R2Tg and R2Bm bind to the same DNA substrate with similar DNA-binding domains, they read out different bases of the target using different residues from the same domains. Since the RUM region contains 4 potential CpG methylation sites, this may underlie the differential sensitivity of R2Tg and R2Bm to DNA methylation (Fig. 1E).

The two additional N-terminal ZnF domains in R2Tg, characteristic of the R2-A clade of R2 elements, were not immediately apparent in our cryo-EM density. However, after low-pass filtering we were able to dock AlphaFold models of these domains above the minor groove of the target DNA between positions −12 to −9 (Fig. 2B). The weak cryo-EM density suggests that in the state of R2 captured on the cryo-EM grid, these domains do not bind the target DNA strongly. However, it is possible that these domains could bind more tightly and/or be involved in target site recognition before bottom strand cleavage, or plausibly after the conformational change that leads to top strand cleavage. Likewise, we speculate that the Myb domain, which we did not observe in the cryo-EM density, may be important for DNA recognition in a different state of the complex. More functional and structural investigation will be required to address these points.

Overall, our structural and biochemical observations suggest important features for an engineered system. Specifically, the synthetic RNA template should minimally contain 3′ homology to the human 28S and a pseudoknotted RNA element in the 3′ UTR to enable TPRT. Further, R2Tg efficiently cleaves the target top strand. This

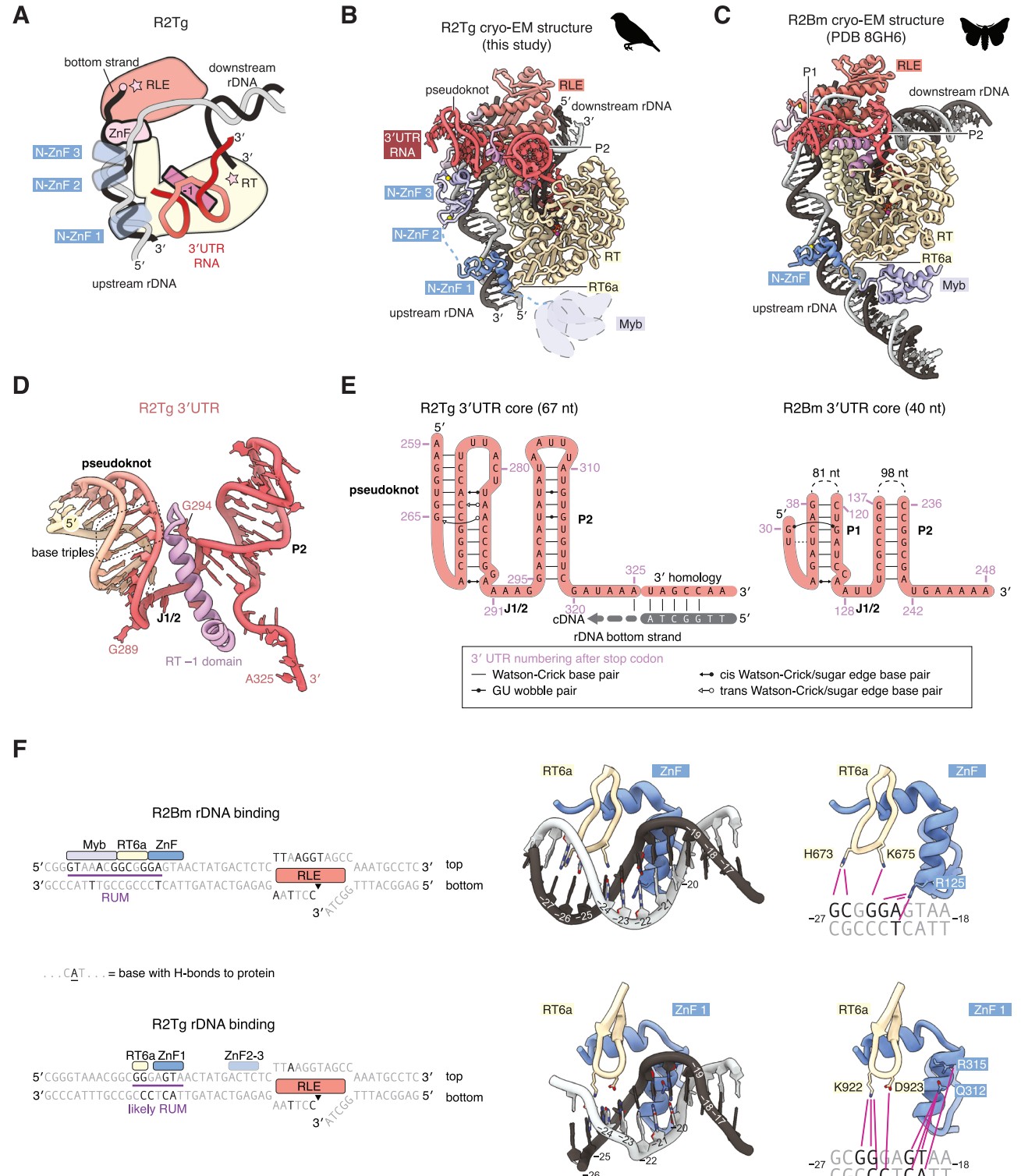

**Fig. 2 | Cryo-EM structure of the R2Tg retrotransposon initiating TPRT.**
**A** Cartoon of the R2Tg cryo-EM structure. Stars represent active sites. **B** Atomic model for the R2Tg TPRT complex. The Myb domain is flexibly attached, indicated by dotted outlines. ZnF, zinc finger; RT, reverse transcriptase; RLE, restriction-like endonuclease. **C** Corresponding view of the previously determined structure of the R2Bm TPRT complex[19]. **D** Structure of the ordered core of the R2Tg 3′ UTR and the associated RT − 1 domain. **E** Secondary structure diagram of the R2Tg 3′ UTR compared to the corresponding core of the R2Bm 3′ UTR. Non-canonical base pairs are shown according to the notation of Leontis and Westhof[96]. **F** 28S rDNA binding by R2Bm compared with R2Tg. Left: bases shown in black form hydrogen bonds to the R2 protein, which may indicate specific recognition of these bases. Right: purple lines indicate the hydrogen bonds in more detail.

suggests that it may not require the 5′ UTR to promote top strand cleavage, as was proposed for R2Bm[16]. However, the protein's ability to create TPRT-independent double-stranded breaks (DSBs) in the absence of functional 3′ UTR also suggests a need to balance protein and template amounts to skew edits toward the reverse transcription-dependent DNA cleavage for desired insertions. With these considerations in mind, we aimed to engineer an optimized all-RNA system for transgene insertion in the human 28S locus.

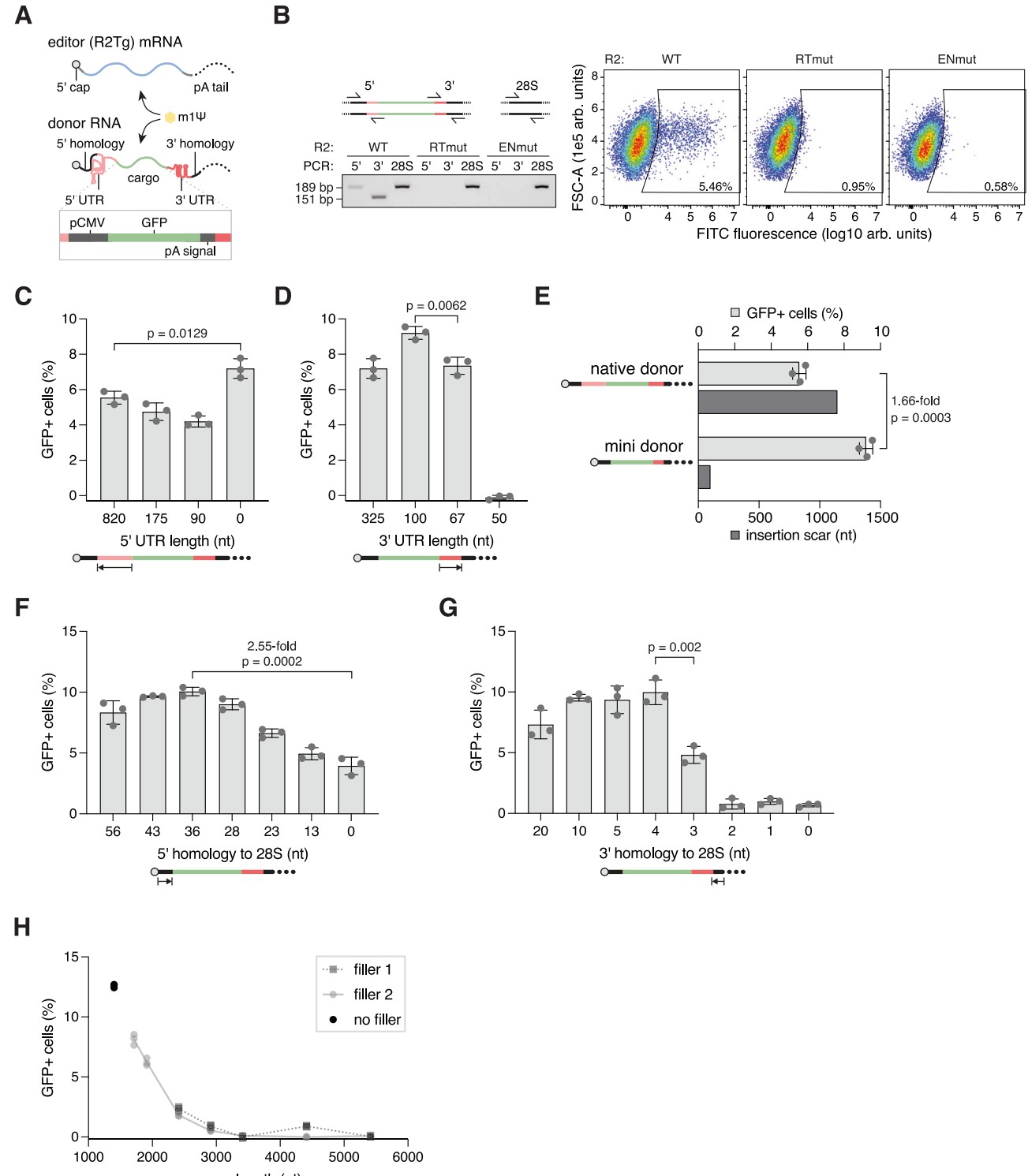

## Engineering a compact RNA system for DNA insertion

To start, we split the native R2Tg system into two RNAs, the editor and the donor (Fig. 3A). For the editor, we used a human codon optimized R2Tg ORF flanked by UTRs that increase mRNA translation and stability[39] (Supplementary Fig. 4A, vaxTg). For the donor, we flanked a GFP reporter cargo with R2Tg UTRs alongside 36 nt of 5′ homology and 5 nt of 3′ homology. Like the native R2 experiment in Fig. 1B, both RNAs have a 5′ cap, m1ψ substitution, and a poly(A) tail. Cotransfecting HEK293FT cells with the donor and editor RNAs resulted in positive junction PCR signals for both the 5′ and 3′ ends, indicating successful cargo integration (Fig. 3B). Flow cytometry of the transfected cells

confirmed that approximately 5% of cells were GFP+ within 24 h of transfection (Fig. 3B). This integration depends on the reverse transcriptase and endonuclease function of R2Tg, as mutating the RT (D878A and D879A) or endonuclease (D1275A and D1288A) domains abolished signal (Fig. 3B).

The RNA donor in the above experiment used the native R2Tg UTRs, whereas our biochemical and structural data show that R2Tg does not require the 5′ UTR for top strand cleavage and only requires the final 65 nucleotides of the 3′ UTR for TPRT in vitro (Figs. 1F, 2E). To test if this compact 3′ element alone is sufficient for functional insertion in mammalian cells, we progressively truncated the 5′ UTR from

**Fig. 3 | R2Tg mediates transgene insertion in human cells. A** Schematic of the functional reporter system. R2Tg mRNA encodes the human codon optimized ORF, and the donor RNA contains a GFP reporter flanked by R2Tg's UTRs. **B** Reporter integration (left) and GFP expression (right) when donors are cotransfected with wildtype R2Tg (WT), reverse transcriptase mutant (RTmut), or endonuclease mutant (ENmut) (repeated two additional times with similar results). "28S" products were diluted 5-fold relative to other products prior to gel electrophoresis. **C** Effects of progressively reducing the donor's 5′ UTR from the 3′ direction ($n = 3$). Each condition was transfected with the same molar quantities of editor and donor RNAs (1:2 molar ratio). Filler RNA compensates for differences in total mass. **D** Effects of progressively reducing the donor's 3′ UTR from the 5′ direction ("325" data in 3c as "0 nt 5′ UTR") ($n = 3$). Each condition was transfected with the same molar quantities of editor and donor RNAs (1:2 molar ratio). Filler RNA compensates for differences in total mass. **E** Comparison of native donor and mini donor's integration efficiencies (data from 3c,d) and theoretical insertion scar size ($n = 3$). **F** Effects of progressively reducing the donor's 5′ homology from the 5′ direction ($n = 3$). **G** Effects of progressively reducing the donor's 3′ homology from the 3′ direction ($n = 3$). **H** Effects of progressively increasing the cargo's length with two different filler sequences ($n = 3$). Each condition was transfected with 400 ng of total RNA with the same molar quantities of editor and donor (1:2 molar ratio). Filler RNA compensates for differences in total mass. For bar graphs, each dot represents a technical replicate, the bars represent the mean, and the error bars represent the standard deviation. When relevant, p values are calculated with Student's two-sided t test. Source data are provided as a Source Data file.

the 3′ end, starting with sequences upstream of the first ZnF domain of R2Tg (Fig. 1C). We then assayed R2Tg's ability to integrate the resulting RNA donors. We found that the 5′ UTR was not required for integration (Fig. 3C and Supplementary Fig. 4B). On donors lacking the 5′ UTR, we progressively truncated the 3′ UTR from the 5′ end and found that R2Tg effectively integrates donors with only 67 nt of the 3′ UTR, but truncating it to 50 nt abolishes integration, in agreement with our biochemical data (Fig. 3D and Supplementary Fig. 4C). Slightly higher integration efficiency was observed with a 100-nt 3′ UTR. In addition, various changes to the P2 sequence that conserved its structure all supported transgene insertion. This includes substituting the 3′ UTR with other avian R2 3′ UTRs[21] (Supplementary Fig. 4D–H).

Based on these results, we created a compact "mini donor" RNA that lacks the 5′ UTR and contains 100 nt of the R2Tg 3′ UTR. Compared to the native design, the mini donor reduces the system's insertion scar by more than 1 kb and increases the integration efficiency by approximately 1.66-fold (Fig. 3E). For applications where it is important to minimize the system payload or non-transgene sequence, the 3′ UTR can be shortened by another 33 nt (to the minimal 67-nt 3′ UTR).

Beyond the UTRs, 28S rRNA sequences usually flank the R2 RNAs[10]. To determine the amount of homology needed for efficient integration, we systematically varied the homology lengths on either side of the donor (Fig. 3F, G). We found that although the efficiency is low, R2Tg integrated donors with no 5′ homology to the target locus. This is possibly achieved through the less precise integration process used by R2 orthologs whose 5′ ribozymes cleave after the 28S sequence, resulting in R2 RNAs with no 5′ 28S homology[10,40–42]. Nevertheless, increasing 5′ homology increased integration efficiency up to 2.55-fold (Fig. 3F), and we proceeded with donors containing at least 28 nt of 5′ homology. By contrast, R2Tg requires at least 4 nt of 3′ homology to efficiently integrate the mini donor (Fig. 3G). This is consistent with our biochemical results that demonstrate R2Tg's need for 3′ homology to efficiently initiate TPRT (Fig. 1D). Under these parameters, our system can integrate cargos up to ~3 kb, although integration efficiency decreases as the cargo size increases (Fig. 3H), with or without the R2Tg 5′ UTR (Supplementary Fig. 4I).

### Chemically modifying donor RNA and optimizing delivery method increased integration efficiency

Because most RNA is relatively short lived in mammalian cells, we reasoned that donor stability limited our system efficiency. Consistent with this, adding a poly(A) tail on the donor RNA improved integration efficiency (Supplementary Fig. 5A)[43]. To further modulate stability while keeping the system compact, we chose to use chemical modifications. We aimed to expand on the m1ψ nucleobase modification that we already used, which produced higher integration efficiency than mini donors that use pseudouridine (ψ) or uridine (U) (Fig. 4A)[44].

Our first clue about which modifications to use came from the surprising observation that R2 5′ UTR lowers donor integration efficiency (Fig. 3C). We hypothesized that this was because the donor's 5′

cap, which would have been cleaved off by the HDV-like ribozyme in the 5′ UTR, protected the RNA from 5′ to 3′ exonuclease degradation (Supplementary Fig. 5B)[45,46]. Indeed, when comparing the donors with and without 5′ caps, we observed that capping only affects integration efficiency in donors without 5′ UTR (Fig. 4B). Thus, mini donor's design may enable us to further modify the 5′ end of the RNA to improve integration efficiency. To test this, we used a 5′ modification containing a locked nucleic acid cap analog with adenine as the +1 base, followed by five nucleotides with 2′ O-methyl (2′OMe) backbone modifications (Fig. 4C)[47]. Compared to the standard Cap 1[28], this modification is more resistant to the human decapping enzyme 2 (hDcp2)[47] and increased integration efficiency by nearly 2-fold (Fig. 4D, modCap). Adding phosphorothioate-end-modified multi-tails (MT)[48] to the 3′ end of the donor RNA also increased the integration efficiency by approximately 1.41-fold (Fig. 4D), but combining the 5′ and 3′ modifications only marginally improved the integration efficiency (Supplementary Fig. 5C), suggesting that the 5′ stability of the donor may be more limiting. Thus, we proceeded with modCap and refer to mini donors using this modification as "miniMod" (Fig. 4E).

In nature, R2 RNA is co-transcribed with the 28S rRNA and lacks the 5′ cap usually found on polymerase II transcripts[10,45]. Instead, the 5′ HDV-like ribozymes of R2 elements likely protect the RNA from degradation[49,50]. To explore this as an alternative approach to enhance integration efficiency, we tested the effects of adding non-native ribozymes to the donor. In total, we examined eight previously identified ribozymes from other R2 orthologs[49], selected for their small sizes and cleavage sites before the 28S sequence. We also tested four non-R2 ribozymes, including a hammerhead ribozyme, which cleaves 3′ of the ribozyme secondary structure and is not expected to confer additional RNA protection (Supplementary Fig. 5D). We found that the ribozyme from the R2 element of European earwig *Forficula auricularia* conferred the best integration efficiency: 2.71-fold and 1.36-fold higher than donors with a hammerhead or Tg ribozyme, respectively (Supplementary Fig. 5D, Fa). However, external factors such as temperature, ion concentrations, and surrounding sequences affect ribozyme folding[51,52]. In the interest of minimizing donor size and maximizing context-independent stability, we continued to use miniMod as our donor. Nevertheless, in agreement with a recent report, these results suggest that HDV-like ribozymes and other stabilizing RNA secondary structures are viable approaches to enhance integration efficiency[50].

We next explored how changes to the RNA delivery method affect integration efficiency, as the method impacts RNA uptake, release, and host response[53,54]. Thus far, we delivered our all-RNA system into mammalian cells through commercial Lipofectamine MessengerMax (MMax). To explore alternative methods, we packaged the mini donor system with non-liposomal transfection reagents (TransIT) and SM-102 lipid nanoparticles (LNPs)[55] (Fig. 4F) (Methods). Strikingly, LNP delivery increased efficiency by more than 4-fold compared to MMax. By combining the miniMod donor with the optimized LNP delivery, we could achieve functional integration in more than 90% of the cells after ~44 h (Fig. 4G–I and Supplementary Fig. 5E).

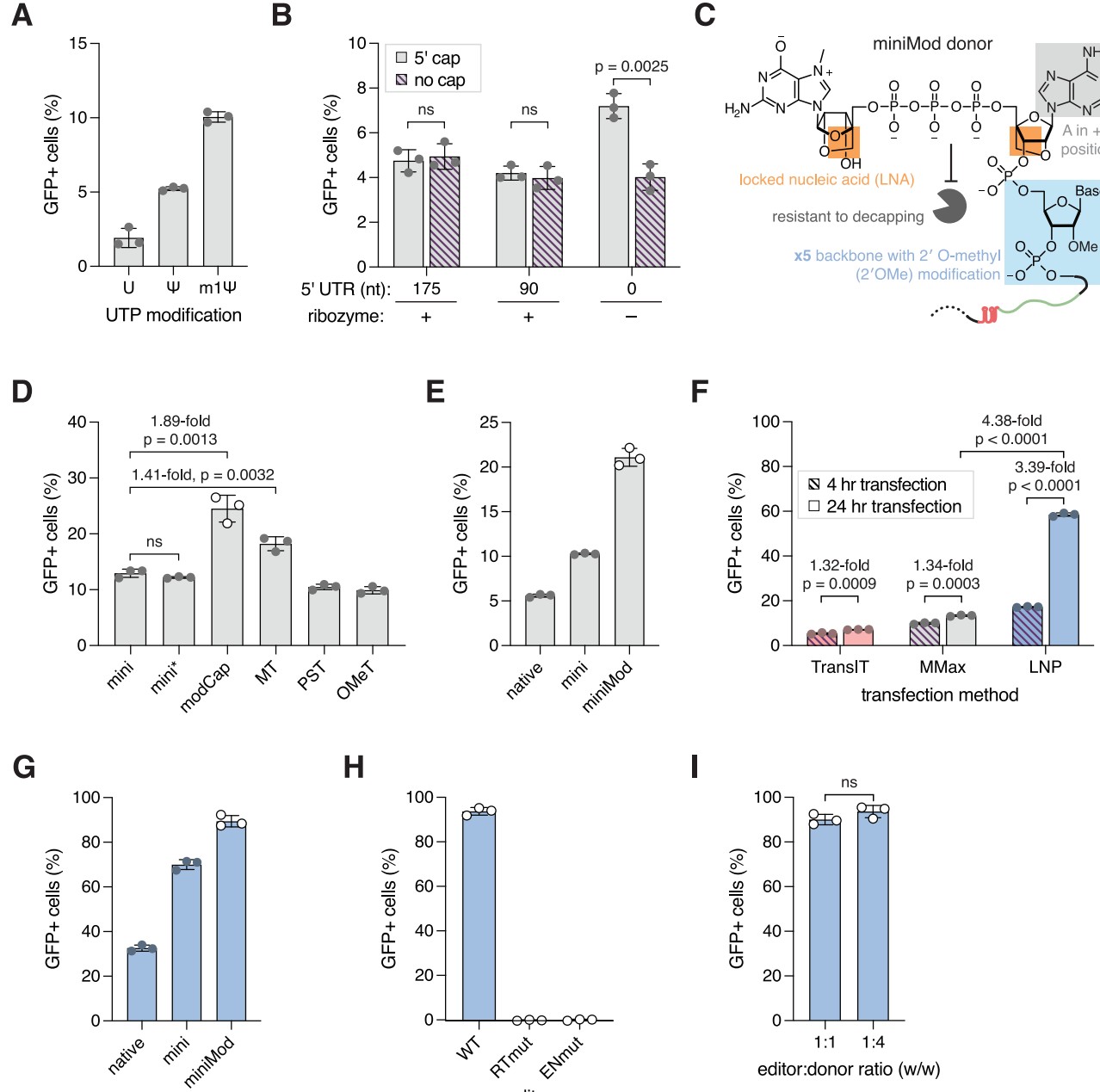

**Fig. 4 | Optimizing donor modifications and system delivery increase integration efficiency. A** Effects of replacing 100% of the uridines with modified uridines in the donor RNA (*n* = 3). U, uridine; ψ, pseudouridine; and m1ψ, N[1]-methylpseudouridine ("m1ψ" data in Fig. 3F as "36 nt homology"). **B** Integration efficiencies of donors with and without 5′ cap analog in the presence and absence of R2Tg's 5′ HDV-like ribozyme ("+cap" data also in Fig. 3C) (*n* = 3). **C** Schematic of miniMod donors' 5′ modification. **D** Effects of ligating chemically modified 5′ or 3′ ends to the donor RNA. mini, mini donor with 36-nt 5′ homology and 30-nt poly(A) tail (*n* = 3); mini*, mini donor with 42-nt 5′ homology and 40-nt poly(A) tail; modCap, mini* with 5′ modification described in 4c; MT, mini* with branched multi-tail; PST, mini* with phosphorothioated poly(A) tail; OMeT, mini* with 2′-O-methyl poly(A) tail. **E** Integration efficiencies of native, mini, and miniMod donors delivered via MMax at 1:1 editor:donor ratio (w/w) (*n* = 3; MMax 24 hr vs. LNP 24 hr p = 6.73e-8; LNP

24 hr vs. LNP 4 hr p = 7.99e-8). **F** Effects of delivering the system with different reagents. Cells were transfected for either 4 or 24 h and analyzed by flow cytometry 24 h after transfection (*n* = 3). TransIT, TransIT-mRNA; MMax, Lipofectamine MessengerMax; LNP, SM-102 lipid nanoparticles. 1:1 editor:donor ratio (w/w) were used for all conditions. **G** Integration efficiencies of native, mini and miniMod donors delivered via LNP at 1:4 editor:donor ratio (w/w) (*n* = 3). **H** Integration efficiency when miniMod is cotransfected with wildtype R2Tg (WT), reverse transcriptase mutant (RTmut), or endonuclease mutant (ENmut) via LNP delivery (*n* = 3). **I** Integration efficiency with different editor:donor ratios (w/w), assayed with miniMod donors and LNP delivery (*n* = 3). For bar graphs, each dot represents a technical replicate, the bars represent the mean, and the error bars represent the standard deviation. When relevant, p values are calculated with Student's two-sided t test. Source data are provided as a Source Data file.

To determine the effect of editor:donor ratio on integration efficiency with LNP delivery, we tested a range of ratios with mini donors. A 1:1 ratio (w/w) produced the highest %GFP+ cells, while adjusting the ratio in either direction gradually decreased efficiency (Supplementary Fig. 6A). To investigate signal persistence on the population level, we

followed transfected cells over 20 days (Supplementary Fig. 6B). Consistent with a recent study, the population of GFP+ cells decreased over time[21]. Samples transfected with high amounts of editor tended to have a sharper decline in %GFP+ cells, while those transfected with low amounts of editor maintained a steady GFP+ population over time

(Supplementary Fig. 6C, left). Similarly, R2Tg variants with lower activity, such as one expressed from mRNA without UTRs or one with an N-terminal deletion (Supplementary Fig. 4A), produced little or no change in %GFP+ cells over time (Supplementary Fig. 6D, left).

Together, these results suggest that high R2Tg activity boosts integration but decreases cell proliferation rates. This may be partly due to more insertion events, as suggested by a previous study[21], but also due to R2Tg's ability to generate TPRT-independent DSBs, as seen in our biochemical assays (Fig. 1F). Consistent with the first reason, we observed that the median fluorescence intensity of the GFP+ population of cells (as a proxy for insertion number)[21] decreased over time in conditions with high R2 activity, suggesting better outgrowth of cells with fewer insertions (Supplementary Fig. 6C, D, right). In a separate experiment, we tracked the divisions of transfected cells with a dye dilution assay. As anticipated, GFP+ cells underwent fewer cell divisions than GFP- cells over the course of four days (Supplementary Figs. 7A, B).

To explore the second reason, we asked if R2Tg also creates TPRT-independent DSBs in mammalian cells. We introduced R2Tg with and without donor RNA and assayed indels at the 28S target. Consistent with in vitro results, the editor cleaved its target even in the absence of donor, leading to approximately 3.46% indel formation but no substantial increase in the γH2AX DNA damage marker[56] (Supplementary Figs. 7C–E). In agreement, R2Tg with and without donor RNA slowed cell proliferation more than its mutant counterparts (Supplementary Figs. 7F, G). This partly explains why in the previous report, an R2 mutant with diminished endonuclease activity produced better transgene stability, albeit at the cost of integration efficiency[21]. It also suggests that other means to control R2Tg activity, such as the incorporation of degron or small-molecule inhibitors[57–59], may improve future iterations of the system.

Importantly, although further studies are still needed to fully disentangle the relative contributions of R2Tg's activities on cell proliferation, these results nevertheless highlight the need to adjust the editor amount or strength to balance integration efficiency and transgene persistence. Since 1:1 and 1:4 editor:donor ratios yield the same integration efficiency with miniMod donors (Fig. 4I), we proceeded with the 1:4 ratio to reduce unnecessary editor load.

## Characterization of R2Tg-mediated integrations

To determine if the engineered all-RNA R2Tg system retains 28S specificity, we performed a modified tagmentation-based tag integration site sequencing (TTISS)[60] assay on the gDNA of cells transfected with LNP-delivered system with miniMod donors, reading out from the 3′ UTR of the donor into the insertion site (Fig. 5A). The resulting reads reveal that R2Tg integration is highly specific. More than 99% of the reads mapped to the 28S or sequences with ≥90% identity to 28S (likely pseudo-28S[61]), and the junction rarely deviated from the expected sequence (Fig. 5B and Supplementary Fig. 8A). The high specificity of R2Tg is consistent with previous reports on similar engineered systems evaluated using different readouts[21,22]. The MMax-delivered system using mini donors also showed high integration specificity with similar 3′ junctions. Interestingly, it has more off-targets than the LNP-delivered miniMod counterpart, potentially a consequence of the higher amount of editor:donor ratio in this condition (1:1 vs. 1:4 for MMax vs. LNP, respectively).

We aligned the rare off-target insertion sites for the MMax-delivered mini donors and found three features (Fig. 5C, Supplementary Data 3). First, the sequences downstream of the insertions strongly matched 4-5 nt of the 3′ homology on the donors, consistent with the 5 bp of primer/template base pairing within the RT active site in the R2Tg structure (Fig. 2E). Second, the sequences 19–25 bp upstream of the insertion site matched the corresponding positions of the on-target 28S target site, with a "T" 20-bp upstream of the insertion showing particularly strong conservation. This target region is bound by the RT6a and ZnF1 domains of R2Tg (Fig. 2F) and suggests these interactions are important for target site determination. Third, 9-bp upstream of the insertion site showed a 5′ T/C-rich motif followed by a more 3′ T/A-rich motif, similar to but not directly alignable to the 28S target sequence. This region partially overlaps the binding site of ZnF2, which was not well-resolved in the cryo-EM density, but suggests ZnF2 may have a role in sequence recognition.

To further characterize the integrations of the LNP-delivered miniMod system, we performed long-read sequencing, enriching for the products with Nanopore Cas9-targeted sequencing (Methods)[62]. Of the completed integrations, 53.4% were full-length, while others had truncations or, more rarely, outcomes such as duplications and snapbacks (Fig. 5D and Supplementary Fig. 8B, C). To calculate the full-length transgene copy number, we performed digital droplet PCR (ddPCR) in parallel. The results suggest an average of $3.8 \pm 0.8$ (mean $\pm$ SD, $n = 3$) 5′ junctions per genome. In the long-read sequencing results, 519 reads contained sequences likely detectable by ddPCR, while 452 reads were classified as full-length integrations. From these ratios, we estimated approximately 3.3 copies of integrated full-length transgenes per genome (3.8*(452/519)).

## The engineered system mediated donor integration in diverse mammalian cell types

Finally, we sought to test our optimized system in a range of mammalian cell types. We independently transfected a panel of mammalian cells with two different LNPs: one co-formed with miniMod donor and R2Tg mRNA and the other carrying H2B-mCherry to serve as a proxy for transfection efficiency (Fig. 5E). We found that our optimized system functions in diverse human and mouse cell lines derived from different tissues, albeit at varying efficiency ranging from 4% in HepG2 cells to over 80% in AC16, HeLa, and HEK293FT (Fig. 5F and Supplementary Fig. 9A). These differences correlate with the LNP transfection efficiency of the cells, as estimated by the median fluorescence intensity (MFI) of samples transfected with the mCherry LNP (Fig. 5G). This suggests that integration efficiency is heavily affected by the RNA delivery efficiency.

We also investigated whether miniMod's modification increases integration efficiency in other cell types. Indeed, compared to cells transfected with mini donor and R2Tg LNPs, miniMod produces higher integration efficiency in all tested cell types, with improvements ranging from 1.33-fold (AC16) to 5.54-fold (BJ) (Supplementary Fig. 9B, C).

In addition to immortalized cell lines, we tested our system in human primary cells and mouse embryonic stem cells. In human skeletal muscle myoblasts (HSMM), LNP-delivered mini donors achieved approximately 15% integration efficiency (Fig. 5H), while miniMods integrated with approximately 0.4% efficiency in human primary T cells (Fig. 5I). In E14 mouse embryonic stem cells, the system delivered via MMax with an H2B-mCherry cotransfection marker integrated with 0.6% efficiency. Considering that RNA delivery may be limiting our efficacy, we also gated on ES cells expressing a high level of the cotransfection marker. In this highly transfected population, the integration efficiency is 1.8% (Fig. 5J). This result further highlights the importance of robust RNA delivery.

We also tested if our system functions in contexts with little to no cell division. Our system successfully integrated nanoluciferase in post-mitotic myotubes that form multinucleated syncytia (Supplementary Fig. 10A)[63]. It also functioned in HEK293FT cells kept in the early S phase by the cell cycle inhibitor aphidicolin[64], albeit with lower efficiency than untreated cells (Supplementary Fig. 10B). Interestingly, cells that were transiently treated with aphidicolin, then released upon transfection had the highest integration efficiency out of the three conditions (Supplementary Fig. 10B, wash). Together, these results suggest that our system does not require active cell division. However, active division boosts its efficiency, potentially due to changes in endogenous repair machineries or chromatin/nuclear accessibilities

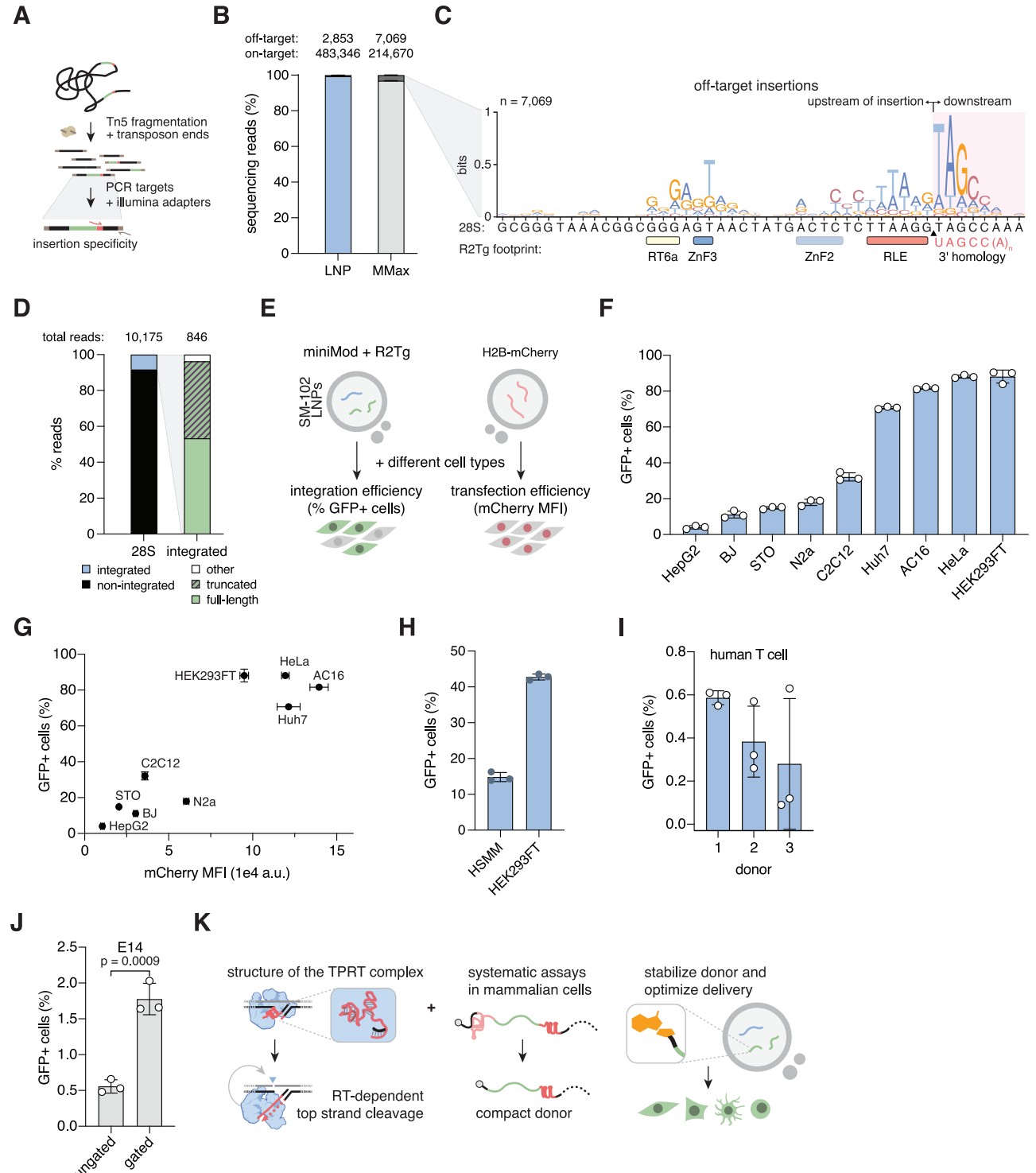

associated with different cell cycle phases[2,65]. Finally, we also demonstrate the ability of our system to integrate different cargos, including an engineered variant of IscB, a programmable RNA-guided DNA endonuclease[66], which led to ~7% indels at the targeted locus upon ωRNA introduction, indicating functional integration (Supplementary Fig. 10C–G).

## Discussion

Here we characterized the R2 element originating from the zebra finch *T. guttata* and rationally engineered it into an all-RNA system for DNA insertion in mammalian cells. Our study provides detailed structural

and biochemical characterization of a member of the R2-A clade, highlighting important differences between the previously characterized R2-D from the silk moth *B. mori*[19]. We showed how the R2Tg protein binds and cleaves its target DNA, and discovered that it recognizes a small pseudoknotted RNA element in the 3' UTR to initiate TPRT. Through systematic testing in mammalian cells, we demonstrated that this element is necessary and sufficient for R2Tg to mediate functional integration in mammalian cells. Further, we showed the importance of donor RNA stability, enhanced it through chemical modifications, optimized delivery, and achieved high integration efficiency across diverse mammalian cell types (Fig. 5K).

**Fig. 5 | R2Tg-mediated insertions in HEK293FT and the system's efficiency in mammalian cells. A** Schematic of the modified TTISS assay to evaluate R2 integration specificity. **B** TTISS reads that map to 28S on- and off-targets. Samples were transfected with LNPs (co-formed with miniMod donor and R2Tg mRNA) or MMax (mini donor and R2Tg mRNA) (n = 3). Off- and on-target counts are the sum of the replicates). **C** Sequence logo calculated for off-target insertions in the MMax samples. Logo was calculated from 7,069 reads mapping to 1530 unique loci, of which 112 loci had more than 10 reads each. X-axis shows the corresponding on-target 28S sequence, with footprints of R2Tg domains annotated based on the cryo-EM structure. **D** Integration products quantified by nanopore Cas9-targeted sequencing. Samples were transfected with LNPs co-formed with miniMod donor and R2Tg mRNA. **E** Experimental design to test the optimized system in a panel of mammalian cells with two sets of transfections. **F** Integration efficiency of our all-RNA system in different mammalian cell types (n = 3). **G** Comparison between integration efficiency and LNP transfection efficiency for each cell type. The latter is

estimated by the MFI of mCherry in the H2B-mCherry transfected cells (n = 3 for each cell type, data presented as mean ± standard deviation). **H** Integration in HSMMs. 96-well samples were transfected with 150 ng of LNP-encapsulated mini donor and editor. Cells were assayed approximately 40 h after transfection. HEK293FT cells transfected with the same conditions were used for comparison (n = 3). **I** Integration in primary human T cells with LNP-encapsulated miniMod and editor (n = 3). **J** Integration in E14 mouse embryonic stem cells with MMax-delivered miniMod, editor, and mCherry cotransfection marker (n = 3). Ungated sample represents all single cells. Gated sample represents the population with top 25 percentile of mCherry fluorescence. **K** Schematic of our key findings. For bar graphs, each dot represents a technical replicate, the bars represent the mean, and the error bars represent the standard deviation. When relevant, p values are calculated with Student's two-sided t test. Source data are provided as a Source Data file.

Recently, two other groups demonstrated R2Tg's ability to insert transgenes in mammalian cells[21,22]. Precise RNA-mediated insertion of transgenes (PRINT) primarily uses R2Tg and R2Za to integrate transgenes flanked by R2 elements from *Tribolium castaneum* (5′ HDV-like ribozyme) and *G. fortis* (3′ UTR), while en-R2Tg modified the R2Tg editor to increase the integration efficiency of donors using 175 and 81-nt of 5′ and 3′ UTRs, respectively. By discovering that R2Tg recognizes a 67-nt 3′ UTR structure to initiate TPRT and mediate transgene insertion (Fig. 2E), our study provides insight into why PRINT is compatible with the similarly structured 3′ UTRs of other avian species[21] (Supplementary Fig. 4), and why en-R2Tg could reduce the donors' 3′ UTR to 81-nt[22] (Fig. 3D).

We do not use any R2 5′ UTR in our mini and miniMod donors. In native R2 systems, the 5′ UTR and its HDV-like ribozyme may promote top-strand cleavage of the target DNA[16], process the R2 element from the rRNA transcript, and confer exonuclease resistance[49,50,67]. Our biochemistry data indicates that the R2Tg ortholog does not require the 5′ UTR to cut either strand of the target DNA (Fig. 1F). Because we synthesized our donors through IVT, we designed them with predetermined amounts of 5′ homology to the 28S target, eliminating a need for ribozyme-mediated processing. Thus, in the context of an engineered donor, the 5′ UTR is likely important for stabilizing the RNA through the secondary structures of the HDV-like ribozyme. This is consistent with our ability to replace the ribozyme with a 5′ cap or non-R2Tg ribozymes (Fig. 4B and Supplementary Fig. 5) and supported by another recent study[50]. Thus, we replaced the ribozyme with a chemically modified 5′ cap to create miniMod. This yielded a more compact system with reduced insertion scars that still performed robustly across a wide range of cell types (Fig. 5). More broadly, without the R2 ribozyme, this donor design also opens the possibility to incorporate other existing and future RNA modifications/motifs on the 5′ end of the RNA, such as those advancing the prime editing field[1,68].

We found that our donor does not require extensive homology to the 28S locus. We observed that 4 nt homology on the donor's 3′ end is sufficient for efficient integration (Fig. 3G). This low requirement is reflected in our structure of TPRT initiation, which captured 5 nt base-pairing with the 28S target (Fig. 2), as well as our off-target analysis (Fig. 5C). It is also consistent with our understanding of R2Bm, which requires no homology for reverse transcription, as well as a number of other characterized reverse transcriptases[27,69,70]. On the other end, we found that up to 36 nt of 5′ homology improves integration efficiency (Fig. 3F). This potentially reflects a feature of natural R2 elements: after cleavage from the rRNA, the HDV-like ribozymes of many R2 orthologs leave up to 36 nt of 28S sequence on the 5′ end of the R2 RNA[49]. With our donor design, increasing the 3′ or 5′ homology further did not improve integration (Fig. 3F, G). Our system's use of relatively short 28S homologies is consistent with PRINT's but not en-R2Tg's designs[21,22].

Interestingly, while our donor RNA performed efficiently with m1ψ (Fig. 4A), different uridine modifications in the donor had divergent effects on integration efficiency across the three systems[21,22]. Further study would be necessary to determine whether the discrepancy is a result of different donor design (for example, interference with ribozyme activity[71]), R2 editor, or cell types used to optimize the system. Nevertheless, m1ψ's low immunogenicity and minimal effects on cell viability[44,72] may outweigh its effects on editing efficiency for downstream application.

Our study highlights several outstanding considerations and questions. First, R2Tg integration efficiency decreases with increasing cargo sizes (Fig. 3H), which could limit its application in some therapeutic contexts. Although the coding sequences of many human proteins are below ~2.5 kb[73,74], large genes that cause common inherited diseases, regulatory elements, and additional application-specific components can push the cargo size above the limit of the system[75]. Future work that increases the cargo capacity of the system will substantially expand its application. This could be achieved, for example, by improving the processivity of editing to reduce truncated integration products (Fig. 5D).

Second, our results suggest that R2Tg-mediated cleavage and insertion disrupt 28S and slow cell proliferation (Supplementary Fig. 7). This presents an interesting conundrum: strengthening the editor activity may increase integration efficiency, but at the cost of increased 28S damage. As a result, transgene-positive cells may initially be abundant, but gradually decline due to compromised viability, undermining long-term stability. Enhancing efficiency while preserving cellular health will be key to advancing R2-based therapeutics. Conceptually, this could involve tuning the system to enable low-copy insertions uniformly across the cell population. Additionally, minimizing unproductive outcomes, such as non-TPRT DNA cleavage and incomplete insertions, may improve efficiency while reducing the burden on target cells.

Retargeting insertion to a different locus may also reduce toxicity. Although the 28S has been successfully used as a safe harbor locus for genome engineering[13,14,15], the ability to reprogram our system would nevertheless avoid disruptions to an essential (albeit redundant) locus, while generally expanding its therapeutic potential. Our previous study with R2Bm demonstrated a proof-of-concept for retargeting in vitro using CRISPR-Cas9[19]. The structural insight from this study also offers ways to approach this rationally: for example, retargeting may be facilitated by modulating the RT6a residues involved in target recognition, and by replacing the Myb and ZnF domains associated with the target DNA with programmable DNA binders like the CRISPR-Cas system.

Third, we show that in the LNP-delivered system, efficient transfection is likely a prerequisite for integration (Fig. 5G). However, MMax also efficiently transfects HEK293FT cells but results in lower integration efficiency. An additional key difference seems to be that over time,

the population of cells with functional integrations continues to increase at a pronounced rate in LNP-transfected samples compared to the MMax and TransIT counterparts (Fig. 4F). In addition, we observed that aphidicolin treatment prior to transfection improves integration efficiency, which suggests that there are some cell cycle dependencies at play (Supplementary Fig. 10B). Based on these results, we postulate that efficient integration requires sufficient quantities of donor RNA and editor protein to be present in the cell at the correct time, and that compared to the other two approaches, LNP provides a steady supply of the RNA over longer periods of time, an effect that may be enhanced by the stabilizing modifications on miniMod donors. If this theory is correct, how can we gradually deliver large quantities of RNA in vivo? What cell-cycle dependent factors or repair machinery are involved? These questions are important to further improve R2 into a versatile tool for genome insertion in all cell types.

Looking ahead, we anticipate that the biochemical, structural, and engineering insights from our study will help guide the future development of R2Tg-based knock-in systems. This all-RNA system is uniquely suited for sensitive primary cells, enabling delivery of both the editor and donor with one LNP without the need for viral vectors, while minimizing the risks of innate immune response, random genomic integration, and prolonged exposure to DNA editors commonly associated with DNA-based systems[7–9,76,77]. These strengths can be leveraged, for example, to introduce chimeric antigen receptors (CAR) into T cells[78]. Our study also highlights the potential of an all-RNA system to leverage the rapid advances in RNA delivery and modification technologies[47,48,77]. As the field of RNA therapeutics continues to evolve, new improvements can be rationally incorporated to broaden the application of our system. To achieve the goal of efficient in vivo transgene insertion, one of the most significant advances will likely come from targeted delivery strategies to introduce large RNA quantities into specific cell types.

## Methods

### Plasmids and cloning
All plasmids used in this study are described in Supplementary Data 1. Plasmids were cloned with standard methods, and sequences were verified by sequencing.

### Production of HRV 3 C protease and anti-GFP Nanobody Sepharose resin
HRV 3 C protease was expressed fused to the C-terminal of a His6-tagged NT* solubility tag derived from spider silk proteins. This expression plasmid, pET-NT*-HRV3CP was a gift from Gottfried Otting (Addgene plasmid # 162795)[79]. The anti-GFP nanobody LaG-41 was subcloned from pET21-pelB-LaG-41, a gift from Michael Rout (Addgene plasmid # 172757)[80], into pET-NT*-HRV3CP to yield plasmid pMW183. Both proteins were expressed in *E. coli* BL21(DE3) (NEB) grown at 37 °C in LB supplemented with 0.5% (w/v) glycerol, 0.2% (w/v) α-lactose monohydrate, 0.05% (w/v) glucose, 2 mM magnesium acetate (auto-induction media), and 50 μg/L kanamycin. The temperature was reduced to 18 °C during mid-log phase, and cells were grown for another 16–20 h.

For HRV 3 C protease, cells were harvested, frozen at −80 °C, and suspended in lysis buffer (50 mM Tris-HCl pH 7.4, 500 mM NaCl, 5% glycerol, 5 mM beta-mercaptoethanol). Cells were lysed by sonication and the cleared lysate was bound to Ni-NTA agarose (Qiagen). The resin was washed with lysis buffer containing 20 mM imidazole then eluted with lysis buffer containing 300 mM imidazole. Protein-containing fractions were dialyzed against 20 mM Tris-HCl pH 7.4, 300 mM NaCl, 1 mM DTT, concentrated to 10 mg/mL, flash-frozen in liquid nitrogen and stored at −80 °C.

For the LaG-41 anti-GFP nanobody, cells were suspended in 4×PBS and lysed by sonication. The cleared lysate was bound to Ni-NTA agarose (Qiagen). The resin was washed with 1×PBS + 500 mM NaCl + 20 mM imidazole then eluted with 1×PBS + 500 mM NaCl + 300 mM imidazole. Protein-containing fractions were pooled and coupled to cyanogen bromide-activated Sepharose 4B (Sigma-Aldrich C9142) according to the manufacturer's instructions. The blocking buffer was 1×PBS + 50 mM glycine. The resin was stored at 4 °C in 1×PBS + 20% ethanol.

### R2Tg protein purification
The R2Tg yeast expression plasmid pMW180 encodes residues 212–1390 of the R2Tg ORF with an N-terminal GFPmut2 – 3 C protease fusion, under the control of a dual GAL-GAPDH promoter previously described[81], with a 2μ origin and TRP1 selectable marker. The plasmid was co-transformed with the centromeric plasmid pRS316, which contains a a URA3 selectable marker, into *Saccharomyces cerevisiae* strain BCY123 (MATα pep4::HIS3 prb1::LEU2 bar1::HIS6 lys2::GAL1/10GAL4 can1 ade2 trp1 ura3 his3 leu23,112). Cells were grown on −URA − TRP selective media with 1% (w/v) raffinose to OD600 = 2 before induction with 2% (w/v) galactose and overnight growth at 30 °C. Cells were harvested and resuspended in one volume of 2×lysis buffer (100 mM HEPES-KOH pH 7.9, 1 M KCl, 4 mM magnesium acetate, 15% glycerol, 0.05% Igepal CA-630, cOmplete protease inhibitor cocktail (Roche)) and frozen dropwise in liquid nitrogen. Frozen cell drops were ground to a powder using a coffee grinder containing powdered dry ice. The powder was thawed and centrifuged at 17,000 g at 4 °C for 30 min. The supernatant was incubated with anti-GFP Sepharose (prepared as described above) for 2 h at 4 °C. The resin was washed with Wash Buffer (20 mM HEPES-KOH pH 7.9, 800 mM KCl, 10% glycerol, 0.01% Igepal CA-630) and then 3 C Buffer (20 mM HEPES-KOH pH 7.9, 500 mM potassium acetate, 10% glycerol, 0.1% Igepal CA-630, 5 mM DTT), before incubation with 0.4 mg of HRV 3 C protease (purified as described above) at 4 °C overnight. The supernatant was collected and the resin washed with further 3 C buffer to collect remaining cleaved R2Tg protein. Protein-containing fractions were pooled and flash-frozen in liquid nitrogen without further concentration. Based on band intensity on an SDS-PAGE gel, the concentration is estimated to be 500 nM.

### AavLEA1 protein purification
The AavLEA1 expression plasmid, pET15b-AavLEA1 was a gift from Claude Férec (Addgene plasmid # 53093)[82]. AavLEA1 was expressed in *E. coli* BL21(DE3) (NEB) grown at 37 °C in Terrific Broth (TB) supplemented with 50 μg/L ampicillin. After reaching saturation, the culture was diluted with 9 volumes of room temperature TB + ampicillin, grown for 1 hr at 18 °C, induced with 0.2 mM IPTG and grown for another 16–20 h. Cells were pelleted and suspended in lysis buffer (50 mM Tris-HCl pH 7.4, 500 mM NaCl, 5% glycerol, 1 mM PMSF). Cells were lysed by sonication and the cleared lysate was bound to Ni-NTA agarose (Qiagen). The resin was washed sequentially with Wash1 buffer (50 mM Tris-HCl pH 7.5, 300 mM NaCl, 20 mM imidazole), Wash2 buffer (50 mM HEPES-KOH pH 7.9, 100 mM potassium acetate, 2 mM magnesium acetate, 1 mM ATP), Wash3 buffer (50 mM Tris-HCl pH 7.5, 1 M NaCl, 20 mM imidazole), Wash1 buffer again, before elution with Wash1 buffer containing 500 mM imidazole. Protein-containing fractions were concentrated and purified by size-exclusion chromatography on a Superdex 75 10/300 column run with 20 mM HEPES-KOH pH 7.9, 300 mM KCl. Protein-containing fractions were pooled, diluted with 1 volume of 20 mM Tris-HCl pH 7.4, and purified on a Resource Q anion exchange chromatography column eluted with a linear gradient from 20 mM Tris-HCl pH 7.5 to 20 mM Tris-HCl pH 7.5 + 1 M NaCl.

### RNA in vitro transcription for biochemical experiments and cryo-EM
Templates for in vitro transcription (IVT) were produced by PCR with a T7 promoter added to the forward primer. Completed PCR reactions were diluted 1/10 in the IVT reaction mixture which contained 4 mM

ATP, 4 mM CTP, 4 mM GTP, 4 mM UTP, 20 mM MgCl2, 40 mM Tris-HCl pH 8.0, 10 mM DTT, 1 mM spermidine, and 85 μg/mL of homemade T7 RNA polymerase, and then incubated at 37 °C for 120 min. The pyrophosphate precipitate was pelleted and removed, and reactions were treated with 1/50 volume of RNase-free DNase I (NEB) at 37 °C for 30 min. RNA was purified using 1.4 volumes of SPRIselect paramagnetic beads (Beckman Coulter) and dissolved in water.

## Preparation of DNA substrates

For biochemical assays, 211-bp DNA targets were prepared by PCR with Phusion Flash polymerase (Thermo Scientific) using a plasmid containing the 28S rRNA gene sequence as a template (pMW27). The forward primer for the top strand had a 5′ fluorescein label and PvuII site (sequence /56-FAM/TTTTTCAGCTGGTTGACGCGATGTGATTTCTG) and the reverse primer for the bottom strand had a 5′ Cy5 label (sequence /5Cy5/TTCCCTTGGCTGTGGTTTCG). PCR products were purified with 1.4 volumes of SPRIselect paramagnetic beads (Beckman Coulter) and dissolved in water.

For cryo-EM sample preparation, a larger amount of 211-bp DNA target was needed. For preparative-scale PCRs, we expressed *Pyrococcus furiosus* DNA polymerase (Pfu) with a C-terminally fused *Sulfolobus solfataricus* Sso7d DNA-binding protein from a pET expression vector in *E. coli* BL21(DE3) (NEB) grown at 37 °C in TB supplemented with 0.5% (w/v) glycerol, 0.2% (w/v) α-lactose monohydrate, 0.05% (w/v) glucose, 2 mM magnesium acetate (autoinduction media), and 50 μg/L kanamycin. The temperature was reduced to 22 °C during mid-log phase, and cells were grown for another 16–20 h. Cells were pelleted and suspended in lysis buffer (50 mM Tris-HCl pH 7.4, 300 mM NaCl, complete protease inhibitor cocktail (Roche)) before lysis in an LM20 microfluidizer. The unclarified lysate was incubated at 80 °C for 30 min before centrifugation at 14000 g for 20 min. The supernatant was diluted with 4 volumes of 20 mM Tris-HCl pH 7.4 and applied to a Mono Q anion exchange chromatography column. The flowthrough was precipitated with 60% saturated ammonium sulfate, redissolved in 100 mM Tris-HCl pH 8, and dialyzed against 40 mM HEPES-KOH pH 7.9, 200 mM KCl, 2 mM DTT, 0.2 mM EDTA. The dialyzed protein was diluted with 1 volume of glycerol, and 0.1% Igepal CA-630 and 0.1% Tween 20 were added, before storage at −20 °C.

The purified polymerase was used in a 10 mL PCR reaction, which contained 30 mM Tris-HCl pH 8.8, 80 mM potassium chloride, 16 mM ammonium sulfate, 1.5 mM magnesium chloride, 0.1% Tween 20, 10 mM arginine, 0.2 mM of each dNTP, 0.5 μM biotinylated forward primer (/5Biosg/TTTTTCAGCTGGTTGACGCGATGTGATTTCTG), 0.5 μM Cy5 labelled reverse primer (/5Cy5/TTCCCTTGGCTGTGGTTTCG), 340 ng of pMW27 plasmid template, and 15 μg Pfu-Sso7d polymerase. The reaction was aliquoted into a 96-well PCR plate and cycled between 98 °C for 10 sec, 60 °C for 20 sec, 72 °C for 15 sec, for 30 cycles total. The wells were pooled, diluted with one volume of 200 mM MES pH 6.5 + 30% ethanol, and applied to a 1 mL Resource Q anion exchange chromatography column using an Akta Purifier system. The column was eluted with a linear gradient from 20 mM MES pH 6.5 + 15% ethanol to 50 mM MES pH 6.5 + 2 M NaCl + 15% ethanol. Fractions containing the PCR product were pooled, ethanol precipitated and dissolved in water to 1 mg/mL final concentration.

Methylated DNA templates were prepared with CpG Methyltransferase (M.SssI, NEB) following manufacturer protocol. Resulting products were purified with SPRIselect paramagnetic beads (Beckman Coulter).

## In vitro TPRT reactions

TPRT reactions contained 20 nM labeled DNA substrate, 1 μM 3′UTR RNA, and 50 nM R2Tg protein in a reaction buffer containing 20 mM HEPES-KOH pH 7.9, 400 mM potassium acetate, 5 mM magnesium acetate, and 25 μM of each dNTP. Reactions were incubated at 37 °C for 30 min and stopped with 1 volume of 2x TBE-urea sample buffer

(90 mM Tris base, 90 mM boric acid, 2 mM EDTA, 12% Ficoll Type 400, 7 M urea, and 0.02% bromophenol blue) supplemented with 1 μg RNase A per reaction. Reactions were boiled at 95 °C for 150 s, placed on ice, and run on a precast 10% acrylamide TBE-Urea gel (Invitrogen) at 400 V for 12–15 min. Gels were visualized using a ChemiDoc (Bio-Rad).

## R2Tg complex formation and purification for cryo-EM

R2Tg complex formation used a 1 mL TPRT reaction containing 400 nM of a 211-bp 28S DNA target with a 5′ Cy5 label on the bottom strand and a 5′ biotinylated top strand, ~400 nM of R2Tg protein, 25 μM of 2′,3′-dideoxythymidine, 1.6 μM of 3′UTR RNA from base 177 onwards + 40 nt 3′ homology, in a reaction buffer containing final concentrations 20 mM HEPES-KOH pH 7.9, 400 mM potassium acetate, 5 mM magnesium acetate, 4 mM DTT, 8% glycerol and 0.08% Igepal CA-630.

The reaction was incubated at 37 °C for 30 min before incubation with Pierce Streptavidin Magnetic Beads (Thermo Scientific 88816) for 1 h at 4 °C. The resin was washed with R2 buffer (20 mM HEPES-KOH pH 7.9, 400 mM potassium acetate, 5 mM magnesium acetate) + 0.1% Igepal CA-630, then R2 buffer (without Igepal CA-630), before eluting at 37 °C for 20 min with R2 buffer containing 2 μL (20 units) PvuII restriction enzyme (NEB). Eluted complexes were concentrated with a 30,000 MWCO Amicon Ultra 0.5 mL centrifugal filter (Millipore-Sigma) to OD260 nm = 16.6, before addition of 0.1 mM dTTP.

For cryo-EM grid preparation, a freshly glow-discharged (90 s at 30 mA) Cu300 R1.2/1.3 holey carbon grid (Quantifoil) was mounted in the chamber of a Vitrobot Mark IV (Thermo Fisher Scientific) maintained at 12 °C and 100% humidity. 9 μL of R2Tg complex was mixed with 1 μL AavLEA1 protein (purified as described above) for protection against the air-water interface[83]. 4 μL of the mixture was applied to the grid before blotting with Ø55 grade 595 filter paper (Ted Pella) for 4 s with force +12 and plunging into liquid ethane.

## Cryo-EM data collection

Cryo-EM data were collected using the Thermo Scientific Titan Krios G3i cryo TEM at MIT.nano using a K3 direct detector (Gatan) operated in super-resolution mode with 2-fold binning, and an energy filter with slit width of 20 eV. Micrographs were collected automatically using EPU in AFIS mode, yielding 10,627 movies at 130,000× magnification with a real pixel size of 0.663 Å, with defocus ranging from −1.5 μm to −3 μm with an exposure time of 1.02 s, fractionated into 40 frames and a flux of 25.7 e-/pix/s giving a total fluence per micrograph of 59.6 e-/Å2.

## Cryo-EM data processing

All cryo-EM data were processed using RELION-5.0[84]. Movies were corrected for motion using the RELION implementation of Motion-Cor2, with 4 × 6 patches and dose-weighting. CTF parameters were estimated using CTFFIND-4.1. Particle picking was first carried out using Topaz with the general model[85], yielding 1,360,697 particles. 2D classification followed by 3D classification (using an ab-initio reference derived from on-the-fly processing using CryoSPARC Live) was used to select 59,774 particles that refined to 4.21 Å resolution. These were used to train a Topaz picking model.

The dataset was then picked again using this new model, yielding 323,700 particles that were subject to 3D classification. Two classes containing 182,304 particles was re-extracted with a 400-pixel box, downscaled to 200 pixels, and refined to 3.90 Å resolution.

Particles were subject to Bayesian polishing, using trained parameters and extracting a 400-pixel box that was re-scaled to 288 pixels. After per-particle defocus refinement, 3D refinement produced a 3.44 Å resolution map. Another round of polishing and refinement yielded a 3.31 Å resolution map. The map was improved using 3D classification without alignment, using a mask around the core, 25 iterations, and a regularization parameter "T" of 8. This yielded a small

subset of 30,612 particles with sharp features, which were refined to 3.27 Å resolution. Anisotropic magnification was then refined, followed by per-particle defocus refinement then refinement of beam tilt and trefoil aberrations. A final refinement with Blush regularization produced an isotropic map at 3.20 Å resolution with features consistent with the estimated resolution. Resolution is reported using the gold-standard Fourier Shell Correlation with 0.143 cutoff.

## Model building

An initial model for the R2Tg ORF was generated using AlphaFold2[86] and was adjusted manually using Coot[87]. The 28S DNA target structure was docked from the structure of R2Bm[19] and adjusted. Coot was used to build the 3' UTR RNA de novo. The model was refined first using ISOLDE[88] then with PHENIX[89], just performing one macro-cycle of global minimization and ADP refinement and skipping local grid searches. Figures were generated using UCSF ChimeraX, developed by the Resource for Biocomputing, Visualization, and Informatics at the University of California, San Francisco, with support from National Institutes of Health R01-GM129325 and the Office of Cyber Infrastructure and Computational Biology, National Institute of Allergy and Infectious Diseases[90].

## R2 tree

98 non-LTR retrotransposons from Repbase annotated as R2 elements were examined for open reading frames. All potential ORFs were translated, aligned with MAFFT, and 23 partial ORFs (N-terminally or C-terminally truncated R2 elements) were excluded. Amino acid sequences from the reverse transcriptase NTE(−1) domain to the RLE domain were extracted and realigned with MAFFT. Gaps were removed from the alignment using trimal, and a phyogenetic tree was generated from the gapless alignment using iqtree with flag -B 1000. The tree was midpoint rooted with ete3 and visualized with iTOL.

## IVT for mammalian cells

DNA templates for IVT were generated by digesting pIVT plasmids with PsiI-v2 (NEB) or by PCR using primers listed in Supplementary Data 1 and NEBNext High-Fidelity 2x Master Mix (NEB). Portions of the donors described in Supplementary Fig. 4F and 5D were subcloned from TCARZ-CMV-GFP-PAmin-GF3-R4A22, a gift from Kathleen Collins (Addgene plasmid # 203768)[21]. PCR products generated from plasmid backbones that contain T7 promoters were additionally digested with DpnI (NEB) for at least 1 h at 37 °C. Products were then purified with QIAQuick columns (Qiagen) or AMPure XP beads (Beckman Coulter) following manufacturer protocols.

IVT reactions were performed with T7 polymerase, buffer, and unmodified ATP, GTP, and CTP supplied in the T7 High Yield RNA Synthesis Kit (NEB). Unless otherwise specified, all RNAs are made with CleanCap AG and N1-Methylpseudo UTP (TriLink Biotechnologies). PseudoUTP and UTP are purchased from TriLink Biotechnologies and obtained from T7 High Yield RNA Synthesis Kit, respectively. Samples with modified UTPs were made with 100% substitution.

After IVT, all samples are treated with DNase I (NEB) or TURBO DNase (Thermo Fisher) for at least 15 min at 37 °C. The resulting RNA were then purified with either the RNA Clean & Concentrator kit (Zymo Research) or Lithium Chloride purification (Invitrogen) following manufacturer protocol. Finally, samples were recovered in UltraPure Distilled Water or 1 mM sodium citrate pH 6.4 buffer (THE RNA Storage Solution, Invitrogen). RNA concentration was determined with Nanodrop 2000c (Thermo Scientific).

## Synthesis of chemically modified RNA

The chemically modified RNAs were synthesized as previously described[47,48], and all oligonucleotides used are outlined in Supplementary Data 1. Briefly, branched oligonucleotides were synthesized by CuAAC from the azide/alkyne modified precursors (IDT) and purified by RP-HPLC (Agilent, PLRP-S). Capped oligonucleotides were synthesized by chemical capping of the corresponding 5' phosphorylated oligonucleotides with corresponding m7GDP-imidazolide followed by HPLC purification. For 3' modification, IVT RNA were ligated with synthetic 5' phosphorylated oligonucleotide (molar ratio 1:25–100) using T4 RNA ligase (Promega, M1051) per manufacturer's protocol and purified by RP-HPLC. For 5' modification, in vitro transcribed RNA (uncapped) were first digested with RppH (NEB, M0356S) to yield 5P-mRNA, which were then ligated to chemically synthesized (and capped) oligonucleotides with T4 RNA ligase, and purified by spin column or RP-HPLC. Ligation efficiency was quantified by RNase H assay as previously described[91] and a second round of ligation is performed in presence of unligated product.

## General mammalian cell culture

HEK293FT cells were purchased from ThermoFisher (R70007). HepG2 (HB-8065), E14 (CRL-1821), N2a (CCL-131), C2C12 (CRL1772), STO (CRL1503), and BJ (CRL-2522) were purchased from ATCC. HSMM cells were purchased from Lonza (CC-2580). HeLa cells were a gift from Paul Blainey's lab. Huh7 cells were purchased from Broad's Genetic Perturbation Platform. AC16 cells were purchased from Sigma (SCC109).

HEK293FT, HepG2, HeLa, Huh7, C2C12, STO, BJ, and N2a lines were cultured in DMEM (Gibco) supplemented with 10% fetal bovine serum (FBS) (Seradigm) and 100 units/mL and 100 μg/mL (1X) of Penicillin-Streptomycin, respectively (Life Technologies).

AC16 were cultured in DMEM/F12 (Gibco) supplemented with 12.5% FBS (Seradigm) and 1X Penicillin-Streptomycin, respectively (Life Technologies).

HSMM were cultured in media and supplements supplied in the SkGM-2 Skeletal Muscle Cell Growth Medium-2 BulletKit (Lonza).

E14 were cultured in GMEM (Sigma) with 10% ES qualified FBS (Gibco), 2 mM L-Glutamine (Gibco), 50 μM β-Mercaptoethanol (Sigma), 1× MEM non-essential amino acids (Gibco), 100 units/mL and 100 μg/mL of Penicillin-Streptomycin (Life Technologies), and 1 unit/uL mouse recombinant LIF (PeproTech).

All of the above cells were grown on tissue-culture treated plates in humidified chambers at 37 °C and 5% $CO_2$. For E14 cells, the plates were additionally coated with EmbryoMax 0.1% Gelatin Solution (EMD Millipore) for at least 15 min at 37 °C prior to use with E14 cells.

To passage, all cells were washed once with PBS (Gibco) and trypsinized with TrypLE Express (Gibco) or Accutase (Stem Cell Technologies) for E14. Cells were counted with the automatic counter Cellometer Auto T4 (Nexcelom Bioscience).

## Mammalian cell transfection

Unless otherwise specified, RNA transfections with Lipofectamine MessengerMax (Thermo Fisher) were carried out on cells plated in 96-well plates following manufacturer protocol, with 200 ng total RNA/well and 1:1 editor:donor RNA ratio (w/w). Briefly, cells were plated 1 day beforehand at a density that would reach approximately 70% confluency upon transfection. For each sample, 5 μL of Opti-MEM (Gibco) and 0.3 μL (or 0.5 μL, for experiments with chemically modified RNA and the donor length test in Fig. 3H) of Lipofectamine MessengerMax were mixed and incubated at room temperature (RT) for 10 min. The resulting solution was added to 200 ng of total RNA diluted in 5 μL of Opti-MEM. The mix was incubated at RT for an additional 5 min before it was gently mixed with 100 μL of media and added to the cells.

Transfections with TransIT-mRNA (Mirus Bio) were carried out following manufacturer protocol. Briefly, for each 96-well sample, 0.18 μL of TransIT-mRNA and 0.18 μL of Boost reagent were diluted in 9 μL of Opti-MEM. The resulting solution was mixed with 200 ng of RNA and incubated at RT for 5 min, before it was gently mixed with 100 μL of media and added to the cells.

## Junction PCR

Genomic DNA of transfected cells were extracted with QuickExtract DNA Extraction Solution (LGC Biosearch Technologies) following manufacturer protocols. Briefly, cell pellets were collected 20–26 h after transfection and resuspended in 50 μL QuickExtract DNA Extraction Solution. Solution was heated in a thermocycler at 65 °C for 15 min, 68 °C for 15 min, and 95 °C for 10 min before cooling for use. Junction PCRs were carried out with primers listed in Supplementary Data 1 and 2X Phusion Flash HiFi Master Mix (Thermo Fisher).

## Flow cytometry

To dissociate for flow cytometry, cells were washed once with PBS (Gibco) and trypsinized with TrypLE Express (Gibco) at 37 °C. Cells were then resuspended in FACS buffer made from PBS with 0.5% BSA (LGC Clinical Diagnostics) and 5 mM EDTA (Invitrogen) and transferred to 96 well non-treated V-bottom plates (Costar).

Flow cytometry was carried out on CytoFLEX S with the CytExpert software (Beckman Coulter). All flow cytometry data were analyzed with FlowJo (BD Biosciences). Gating strategies are described in Supplementary Fig. 11.

Unless otherwise specified, cells transfected with Lipofectamine MessengerMax were flowed approximately 24 h after transfection, and cells transfected with lipid nanoparticles (LNP) were flowed approximately 44 h after transfection.

## LNP formulation

Unless otherwise specified, the LNPs used in this study were formulated by hand. Lipid mix was prepared following a previously described recipe[55] with the following reagents (% in molar ratio) diluted in ethanol: 10% DSPC (Avanti Polar Lipids), 38.5% Cholesterol (Sigma), 1.5% DMG-PEG 2000 (Avanti Polar Lipids), and 50% SM-102 (BroadPharm).

RNAs for each transfection condition were premixed and diluted to 0.19–0.20 μg/μL in 10 mM citrate buffer (pH 4.5) to form the aqueous component, while the lipid mix was diluted to 14.9 mM in ethanol to form the organic component. The aqueous component was rapidly added to the organic component in a ratio of 3:1 (v/v) and vigorously mixed by pipetting up and down for 30 s. The resulting solution was immediately diluted in 5 volumes of PBS. The LNPs were stored at 4 °C prior to quantification and use.

LNPs used in Supplementary Fig. 10F were formulated with NanoAssemblr Spark (Precision Nanosystems). Lipid mix was prepared as described above. RNAs for each transfection condition were premixed and diluted to 0.57 μg/μL in 10 mM citrate buffer (pH 4) to form the aqueous component, while the lipid mix was diluted to 29.9 mM in ethanol to form the organic component. After formulation with an aqueous:organic ratio of 2:1 (v/v), the LNPs were diluted in PBS, then buffer exchanged and concentrated using Amicon Ultra-4 Centrifugal Units, 100 kDa MWCO (MilliporeSigma).

Quant-it RiboGreen RNA assay kit (Thermo Fisher) was used to quantify the concentration of encapsulated RNA and the encapsulation efficiency. Results were read out on a BioTek Synergy Neo2 plate reader (Agilent Technologies). Unless otherwise specified, 200 ng of encapsulated RNA were used for each 96-well transfection.

## ddPCR

gDNA extracted with DNeasy Blood & Tissue Kit were digested with EcoRI-HF and HindIII-HF (NEB). 22 μL of ddPCR reaction mix was prepared by combining 11 μL of ddPCR Supermix for Probes (no dUTP) (Bio-Rad), primers for target and RPP30 reference gene (900 nM final concentration each), probes for target and reference gene (250 nM final concentration each), and digested gDNA (1.2 ng/μL final concentration). Droplets were generated with Automated Droplet Generation Oil for Probes (Bio-Rad) using the Automated Droplet Generator (Bio-Rad). The following thermocycling

conditions were used: 95 °C for 10 min, [94 °C for 30 s, 58 °C for 1min] for 40 cycles, and 98 °C for 10 min, 4 °C hold. The resulting products were read on a QX200 Droplet Reader (Bio-Rad). Concentrations were determined using QuantaSoft, and the copy numbers per genome were calculated by dividing the concentration of the target with the concentration of RPP30, assuming two copies of RPP30 per genome.

## Assessing R2Tg target specificity

For the MMax samples described in Fig. 5B, C and Supplementary Fig. 8A, 2.2e5 cells/well of HEK293FT were plated on a 12-well plate one day before transfection. A total of 2200 ng total RNA composed of premixed editor and donor at 1:1 ratio (w/w) were then transfected with 3.3 μL of Lipofectamine MessengerMax following the procedures described above. Approximately 24 h later, cells were harvested for gDNA extraction with the DNeasy Blood & Tissue Kit (Qiagen).

For the LNP samples described in Fig. 5B and Supplementary Fig. 8A, 1.2e5/well of HEK293FT were plated in a 24-well plate one day before transfection. LNPs with 1200 ng encapsulated RNA formulated from premixed editor and donor at 1:4 ratio (w/w) were added to the cells. Approximately 44 h later, cells were harvested for gDNA extraction with the DNeasy Blood & Tissue Kit.

Donor integration sites in the genomes were determined using tagmentation-based tag integration site sequencing (TTISS), as previously described[60] with minor modifications as follows. Briefly, 2 μg of genomic DNA from each sample was mixed with 1 μL of purified Tn5 enzyme loaded with a Tn5 adaptor (5′-CTGTCTCTTATACA-CATCTCCGAGCCCACGAGAC-3′) as described. Tagmented samples were purified using a QiaQuick DNA purification kit (Qiagen) and amplified twice using KOD Hot Start 2× PCR Master Mix (MilliporeSigma) with primers 5′-GCCACCTTTACTTAACCCGGAAAAG-3′ and 5′-GTCTCGTGGGCTCGGAGATGTGTATAAGAGACAG-3′ for 12 cycles and an annealing temperature of 60 °C in the first round of PCR, then with primers 5′-AATGATACGGCGACCACCGAGATCTA-CACTATAGCCTACACTCTTTCCCTACACGACGCTCTTCCGATCTGG AACATATATAATTTATGTGTGTTCGATA-3′ and 5′-CAAGCAGAAGACG GCATACGAGATNNNNNNNNGTCTCGTGGGCTCGGAGATGTGT-3′ (NNNNNNNN refers to barcode sequence) for 20 cycles and an annealing temperature of 62 °C in the second round of PCR. The libraries were subjected to sequencing on an Illumina MiSeq.

To determine the rates of on- and off-target integrations, we used cutadapt to extract read pairs where the start of read 1 matched the R2Tg 3′ UTR. After trimming the R2Tg 3′ UTR sequence from read 1, the read pairs were first mapped to the 28S rDNA sequence using bowtie2. All mapped reads are counted as on-target. Unmapped read pairs were then mapped to the human genome (T2T) using bowtie2. The mapping sites of properly paired reads in some cases corresponded to "pseudo-28S" sequences. Any sites where within a 120 bp window around the insertion site, at least 108 bases matched the wild-type 28S sequence, were also counted as on-target. The remaining sites are counted as off-target reads. For the MMax-delivered mini donor samples, the off-target reads are used to calculate a sequence logo, with each site weighted by the number of mapped reads.

CRISPResso2 was used to determine the nucleotide frequencies at the integration junction[92]. Single-end read 1 results were mapped to 59 nucleotides of the junction sequence, assuming perfect integration (i.e. the "amplicon", GGAACATATATAATTTATGTGTGTT CGATAAATAGCCAAATGCCTCGTCATCTAATTAG). A hypothetical sgRNA sequence was used to specify the junction location (TTA TGTGTGTTCGATAAA). We further specified a minimum homology of 60% to the amplicon and a quantification window of 5 nucleotides centered around position 0 of the sgRNA (i.e. the junction location), filtering out reads with average and single base read quality under phred33 score of 20.

 

## Nanopore long-read sequencing

6e5/well of HEK293FT were plated in a 6-well plate one day before transfection. LNPs with 6000 ng encapsulated RNA formulated from premixed editor and donor at 1:4 ratio (w/w) were added to the cells. Approximately 44 h later, cells were harvested for gDNA extraction with the DNeasy Blood & Tissue Kit. The sequencing library was then prepared according to the Oxford Nanopore's Cas9 enrichment protocol (SQK-LSK109) and its associated publication[93]. To enrich for the target locus, crRNAs (IDT) were designed to target conserved regions within the 28S region, flanking the R2 integration site by approximately 4 kb on each side (sequences in Supplementary Data 1). crRNAs were resuspended at 100 μM and pooled in equimolar amounts prior to mixing with 10 μM of tracrRNA (IDT) and heated at 95 °C for 5 min, then cooled at RT for 5 min. Ribonucleoprotein complexes (RNPs) were assembled by combining 30 pmol of gRNA duplex with 15 pmol of Cas9 in CutSmart Buffer (NEB) and water. The mixture was incubated at RT for 30 min and then stored at 4 °C for up to 2 days before use.

3 μg of input DNA was dephosphorylated with 3 μL of Quick CIP (NEB) at 37 °C for 60 min, followed by heat inactivation at 80 °C for 2 min. Once the sample returned to RT, 15 μL of Cas9-gRNA complex was added alongside 1 μL of 10 mM dATP (NEB) and 1 μL of Taq DNA polymerase (NEB). The mixture was incubated at 37 °C for 45 min and 72 °C for 5 min. The samples were purified with AMPure XP beads (1:1 v/v) and eluted in water. 750 ng of DNA in 22.5 μL of water was mixed with 2.5 μL of Native Barcode (Oxford Nanopore Technologies) and 25 μL of Blunt/TA Ligase Master Mix (NEB) and incubated at RT for 15 min. Barcoded samples were purified once more with AMPure XP beads and the concentrations were determined by Qubit dsDNA HS Assay (Invitrogen), prior to combining them into a 1050 ng total DNA pool.

Oxford Nanopore Ligation Sequencing Kit (v14 chemistry, Oxford Nanopore Technologies) was used to ligate sequencing adapters with Quick Ligase (NEB) for 15 min at RT. Samples were purified with AMPure XP beads (1:0.3 DNA:bead v/v), followed by two washes in the long-fragment buffer (Oxford Nanopore Technologies). Finally, 37.5 μL of sequencing buffer (Oxford Nanopore Technologies) and 25.5 μL of Library beads (Oxford Nanopore Technologies) were added to prepare the final sequencing library. Sequencing was performed on MinION flow cells (v.10.4.1) using the MK1C device.

## Long-read analysis

The full analysis pipeline is available on https://github.com/Clarissa-Schaefer/R2-nanopore-pipeline. Briefly, Basecalling, adapter trimming, and demultiplexing were performed using Dorado (Oxford Nanopore Technologies) in "superior accuracy" mode (dorado-0.9.0-linux-x64 and model v5), followed by quality trimming with cutadapt. Using minimap2, reads were aligned in splice mode to WT and transgene-integrated 28S sequences as references. Reads that aligned to either reference were used for downstream analyses.

FASTA format of the reads were annotated with RepeatMasker to identify the features: 3end_400nt, 3end_200nt (corresponding to 200–400 nt and 0–200 nt downstream of R2Tg's first strand nick site, respectively), 3' UTR, transgene, 5end_200nt, and 5end_400nt (the last two corresponding to 0–200 nt and 200–400 nt upstream of R2Tg's nick site, respectively). Reads that contained ≥ 100 nt of both the 3end and 5end features in the same orientation were used for downstream analysis. Reads with either 3' UTR or transgene annotations were classified as "inserted" and reads without either were classified as "uninserted". "Inserted" reads were trimmed to the regions flanked by the 3 and 5end features. For rare reads containing multiple instances of 3 and 5end features, we identified the same-orientation pair that is located furthest apart on the read and

used the intervening sequence for analysis. All reads were classified into "full-length", "truncated", and "other".

"Full-length" reads contained only one transgene and 3' UTR feature, and deletions/non-aligned bases (D/N in CIGAR) did not occur in ≥4 consecutive bases (accounting for long-read sequencing errors). The "truncated" category is subdivided into "FivePrimeTruncations" and "Jumps". "FivePrimeTruncations" reads contained only one transgene and 3' UTR feature, with deletions/non-alignment in ≥4 consecutive bases, and do not align to the 5' end of the transgene. "Jumps" reads contained either >1 transgene or 3' UTR features that did not overlap on the reference, or a single transgene feature aligning within the first 10 nt of the 5' end of the transgene. Reads that do not belong in "full-length" or "truncated" categories were placed in "other".

For visualization, "full-length" and "truncated" were plotted by overlapping their primary and supplementary alignments in minimap2 with –end bonus 1500 and mismatching penalty -B3. "Other" were visualized based on their feature annotations. Reads that contain the ddPCR probe binding site, flanked by primer binding sites in the correct orientation (permitting ≤2-nt mismatches), were counted as reads that contain a sequence likely detectable by ddPCR.

## Primary T cell isolation

Primary human T cells were isolated from buffy coats obtained from Blood Research Components. Initially, buffy coats were diluted with 1 volume of dilution buffer (PBS, 0.1% BSA, 2 mM EDTA) and used for density centrifugation (Ficoll-Paque Premium 1.084, Cytiva; centrifugation at 600 x g for 30 min without deceleration) in 50 mL Leucosep tubes (Greiner Bio, 227290 P). Cells were aspirated, washed in dilution buffer, and pelleted by centrifugation (500 x g, 5 min). Red blood cells were lysed using ACK lysis buffer (Gibco, A1049201) for 1 min at room temperature, the reaction quenched with PBS and cells harvested by centrifugation (500 x g, 5 min). T cells were isolated using a Pan T cell isolation kit (Miltenyi Biotech, 130-096-535), following the manufacturer's instructions. Briefly, cells were resuspended at 10e6 cells per 40 μl in MACS buffer (PBS, 0.5% BSA, 2 mM EDTA) and labeled with 10 μl of Pan T cell Biotin-Antibody Cocktail. After a 5 min incubation at 4 °C another 30 μl of MACS buffer were added, followed by 20 μl of Pan T Cell MicroBead Cocktail. The cells were incubated for 10 min on ice and subsequently used for magnetic isolation on an LS column (Miltenyi Biotech, 130-042-401). The flow through cell suspension, containing enriched T cells, was collected, counted and aliquots of 5e6 cells were frozen in freezing media (RPMI, 20% FBS, 10% EDTA) until used.

## Primary T cell culture and transfection

Primary human T cells were cultured in Immunocult-XF T cell expansion medium (STEMCELL Technologies, 10981) supplemented with 100 ng/ml rhIL-2 (Peprotech, 200-02-500ug) and activated using T cell activator beads (Thermo Fisher Scientific, 11131D) at 1e6 cells/mL. Three days post activation, cells were enumerated, and the activation status of the cells analyzed by anti-CD25 staining and flow cytometry, at which point more than 99% were CD25 + . T cells were subsequently diluted to a concentration of 0.5e6 cells/ml and 1 μg/ml of ApoE (Peprotech, 1001149) was added to the medium. Cells were distributed into 96-well plates (100 μl) per well and treated with the indicated LNP formulations (200 ng encapsulated mRNA per 5e4 cells). LNPs were coformed with editor and miniMod donor (1:4 ratio, w/w) using the GenVoy-ILM T cell kit for mRNA (Cytiva, 1000683) following manufacturer protocol. Two days after addition of the LNPs, 100 μl fresh T cell media (containing 100 ng/ml rhIL-2) were added to the cells, and the cells analyzed by flow cytometry three days after the treatment.

## Flow cytometry of primary T cells

Cells were de-beaded and harvested by centrifugation (500 × g, 5 min). Media was removed and cells were resuspended in 100 μl Flow Buffer

(PBS, 1% FBS, 2 mM EDTA). For antibody staining, cells were subsequently incubated with 2 μl per well of anti-CD3 (Biolegend, 300328) or anti-CD25 (Biolegend, 302617) for 45 min. Afterwards, the cells were washed twice more in 100 μl of Flow Buffer. The last wash was performed with Flow Buffer containing 100 ng/ml DAPI (Thermo Fisher Scientific, D1306). Finally, cells were resuspended in 100 μl of Flow Buffer and analyzed on a Cytoflex S flow cytometer.

### CellTrace assay

Cells were trypsinized approximately 44 h after LNP transfection and stained with CellTrace Far Red Cell Proliferation Kit for flow cytometry (Invitrogen), following manufacturer recommendations for suspension cells. A subset of the stained cells was immediately analyzed by flow cytometry (d2 samples), while another subset was replated and expanded for an additional 4 days before analysis (d6 samples). Relative cell divisions were estimated by the median fluorescence intensities of the dye at d2/d6.

### Indel analysis

gDNA of transfected cells were extracted with QuickExtract DNA Extraction Solution (LGC Biosearch Technologies). Target sites were amplified with PCR primers containing adaptors for 15 cycles, then with barcoded primers for 15 cycles. Amplicons were sequenced on Illumina MiSeq, and indel frequency and profiles were generated with CRISPresso2, filtering out reads with average quality under phred33 score of 30.

### γH2AX staining

Cells were fixed and permeabilized with BD Cytofix/Cytoperm Fixation/Permeabilization Solution Kit (BD Biosciences) and stained for γH2AX with BD Pharmingen Alexa Fluor 647 Mouse anti-H2AX (pS139) (BD Biosciences), at 1:10 dilution, following manufacturer recommendations. Cells were then analyzed for median fluorescence intensity using flow cytometry. As positive control for DSB damage response, cells were treated with 50 μM of etoposide (MedKoo) for 7 h prior to fixation and staining.

### C2C12 differentiation

C2C12 were differentiated as previously described[94]. Briefly, cells were plated in 96-well plates at 7e3 cells per well the day before induction of differentiation. The next day, media was exchanged to DMEM with 2% horse serum (Gibco, 16050114) and 1× Penicillin-Streptomycin. Media was exchanged every two days. Eight days after induction, cells were transfected with LNP and returned to DMEM with 10% FBS and 1X Penicillin-Streptomycin.

### Luminescence readout

For cells transfected with nanoluciferase donors, luminescence was measured after lysing the cells with Nano-Glo Luciferase Assay System (Promega) following manufacturer protocol. Resulting luminescence was read out on a white 96 well flat bottom assay plate (Corning) with a BioTek Synergy Neo2 plate reader (Agilent Technologies).

### IscB activity assay

Cells were transfected with LNP-delivered IscB miniMod donors and WT or RTmut editors for 2 days. They were then expanded for an additional 2 days prior to ωRNA transfection. Plasmid for ωRNA expression was transfected with Lipofectamine 3000 Reagent (Invitrogen) following manufacturer protocol. Briefly, 200 ng of plasmid was mixed with 0.4 μL of P3000 in 5 μL of Opti-MEM. The solution was then mixed with 0.3 μL of Lipofectamine 3000 diluted in 5 μL of Opti-MEM. The resulting transfection mix was incubated at RT for 10 min before it was gently mixed with 100 μL of media and added to the cells. Cells were harvested 3 days later for indel analysis.

### Statistics and reproducibility

No statistical method was used to predetermine sample size. No data were excluded from the analyses. The experiments were not randomized. The Investigators were not blinded to allocation during experiments and outcome assessment. Statistical tests were performed with GraphPad Prism (v10.4.2).

### Reporting summary

Further information on research design is available in the Nature Portfolio Reporting Summary linked to this article.

## Data availability

Supplementary Data 1 contains all plasmid, oligonucleotide, and crRNA sequences used in this study. The cryo-EM map has been deposited in the Electron Microscopy Data Bank with accession code EMD-47091. The coordinates for the atomic model have been deposited in the Protein Data Bank under accession code 9DOU. The next-generation sequencing dataset for TTISS and nanopore long-read sequencing dataset are available on SRA under BioProject PRJNA1163071. Other data supporting the findings are within the Article and its Supplementary Information. Source data are provided with this paper.

## Code availability

Code for nanopore long-read sequencing analysis is publicly available at https://github.com/Clarissa-Schaefer/R2-nanopore-pipeline and on Zenodo[95] under the MIT license.

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

## Acknowledgements

We thank K. Yuan, J. Strecker, L. Evgeniou, S. Hirano, and D. Li for valuable scientific discussions and K. Kappel, M. Saito, P. Xu, N. Quinones-Olvera, and P. Kumar for critical feedback on the manuscript. We thank the entire Zhang lab for their support and advice. This work was supported by the Helen Hay Whitney Foundation (K.K.E. and M.E.W.), Howard Hughes Medical Institute (K.K.E., M.E.W., and F.Z.), Ruth L. Kirschstein National Research Service Award Predoctoral Fellowship (1F31CA275339-01) from the National Cancer Institute (B.L.), German Academic Scholarship Foundation (C.C.S.), The Gruss Lipper Postdoctoral Fellowship (S.Z.T.), The Outstanding Israeli Female Postdoctoral Program Scholarship (S.Z.T.), Research Science Institute

(N.A.V.), E. Scolnick Professorship (X.W.), Ono Pharma Breakthrough Science Initiative Award (X.W.), Merkin Institute Fellowship, Klarman Cell Observatory (X.W.), Packard Fellowship (X.W.), Sloan Research Fellowship (X.W.), NIH DP2 New Innovator Award (1DP2GM146245) (X.W.), The Yang-Tan Collective (F.Z.), Howard Hughes Medical Institute (F.Z.), The Poitras Center for Psychiatric Disorders Research at MIT (F.Z.), Broad Institute Programmable Therapeutics Gift Donors (F.Z.), The Pershing Square Foundation, William Ackman, and Neri Oxman (F.Z.), and The Asness Family Foundation (F.Z.). Organism silhouettes are from Phylo-Pic.org. Model figures were generated with UCSF ChimeraX, developed by the Resource for Biocomputing, Visualization, and Informatics at the University of California, San Francisco, with support from National Institutes of Health R01-GM129325 and the Office of Cyber Infrastructure and Computational Biology, National Institute of Allergy and Infectious Diseases.

## Author contributions

K.K.E., M.E.W., and F.Z. conceived of the project. K.K.E., M.E.W., D.S., B.L., C.C.S., S.Z., S.Z.T., A.L., M.L.W., C.J.F., and N.A.V. designed and performed experiments. H.C. and D.L. synthesized modified donor RNAs with supervision from X.W. F.Z. supervised the research and experimental design with support from R.K.M. K.K.E., M.E.W., R.K.M., and F.Z. wrote the manuscript with input from all authors.

## Competing interests

F.Z. is a scientific advisor and cofounder of Beam Therapeutics, Pairwise Plants, Arbor Biotechnologies, Aera Therapeutics, and Moonwalk Biosciences. F.Z. is a scientific advisor for Octant. X.W., H.C., and D.L. have submitted a patent application to the U.S. Patent and Trademark Office related to 5′ chemically modified RNAs (International Publication No. WO 2025/04308; PCT converted). X.W. is a consultant, equity holder and scientific cofounder of Stellaromics and Convergence Bio. The remaining authors declare no competing interests.
