## [Transparent Peer Review file · Nature Communications]

Structure and biochemistry-guided engineering of an all-RNA system for DNA insertion with R2 retrotransposons

Corresponding Author: Professor Feng Zhang

Version 0:

Reviewer comments:

Reviewer #1

(Remarks to the Author)

In this manuscript, the authors identify an avian R2 retrotransposon (R2Tg) as active in human cells and provide a comprehensive characterization through in vitro biochemical assays and cryo-EM structure determination. The study reveals interactions of the enzyme with template RNA and target DNA, highlighting notable differences from the previously studied R2 element from *Bombyx mori* (R2Bm). Taking advantage of this structural information, the authors designed a minimal vector system with high activity in human cells. They further optimize the donor RNA structure and chemistry through base modifications, enhancing cellular stability. Combined with refined transfection protocols, this approach achieves remarkably high integration levels, reaching up to 90% under specific conditions and cell types. The manuscript presents a well-executed and clearly described body of work, with meticulously designed and controlled experiments. While R2Tg has been recently identified and repurposed for transgene integration in other studies, including all-RNA-based systems, this work offers significant new insights beyond incremental improvements (such as the cryo-EM structure of the complex, and levels of integration far above what was obtained so-far). A few important questions raised by their findings remain pending and would need to be answered before publication:

1. Off-targets and undesired events: The manuscript lacks detailed characterization of undesired events, such as indels at on-targets resulting from cleavage without insertion and subsequent NHEJ repair, truncations, junction accuracy, and potential inversions. A more comprehensive analysis of these aspects is necessary.
2. DNA damage: Does R2Tg induce detectable DNA damage, such as gamma-H2AX foci? If so, are they primarily localized to the rDNA locus? Do certain conditions (e.g., when top-strand cleavage becomes RT-independent) alter the number or location of these foci?
3. Copy number: What is the average and range of transgene copy numbers under optimized conditions, given the multi-copy nature of rDNA?
4. Transgene size: What is the size limit of the donor RNA? The study only tests GFP as a transgene. How does the system perform with more relevant genes of interest, and what is the maximum size that can be accommodated while avoiding truncations?

Other points:

5. The ZnF order in Fig. 2A appears to be inverted. Please verify and correct if necessary.
6. The transfection of the TTISS donor oligo used to map off-target cleavage sites is not mentioned in the methods section. Please provide clarification.
7. PDB files should be made accessible to reviewers to enable proper evaluation of the new structure and comparison with previous ones.

Reviewer #2

(Remarks to the Author)

Edmonds et al. performed biochemical and structural characterization of the R2 retrotransposon system from *Taeniopygia guttata* (R2Tg). Leveraging the insights from these experiments, the authors were able to rationally engineer a compact R2Tg tool that perform more robustly than the previous native and engineered ones (as reported in Ref 19 and Ref 20, respectively) in multiple types of mammalian cells. Notably, this new system features chemical modifications as well as

truncation on the donor/template RNA, while the R2-encoded “editor” protein remains intact. This work is of significance to the genome engineering field as it offers an optimized 2-RNA system for efficient, targeted gene insertion. The soundness of this research might be further enhanced if the authors could address the concerns described below.

Major concerns:

1. The authors have the 5'-UTR removed from the donor RNA to obtain a “mini donor” while keeping tens of nucleotides 5' homology. This means that there is no functional ribozyme within the 5' end of the new donor RNA. However, the recent study in Ref 20 suggests that the presence of this HDV-like ribozyme is important for R2Tg retrotransposition activity. Moreover, Ref 20 shows the necessity of at least 100 nucleotides (nt) of homology to the target locus at both the 5' and 3' ends. In the current manuscript and Ref 19, the authors demonstrate a few nts (~4) of 3' homology are sufficient for R2Tg protein-mediated integration. The apparent discrepancies between these studies on the same systems need to be clarified.
2. The authors applied ψ and m1 ψ modifications to the donor RNA uridines and found that the m1 ψ modification produced higher integration efficiency than using pseudouridine (ψ) or the native uridine. It is interesting to note that an opposite trend might even be observed with different cell types, as described in Ref 20. A reasonable discussion on this point is recommended.
3. It could be necessary to perform side-by-side quantitative evaluation of the new version of R2Tg with that of e.g., en-R2Tg reported in Ref 20 under same assays conditions in terms of integration efficiency and specificity. Yet, the authors can make at least a simple semi-quantitative comparison from those similar assays conducted in the studies, if present.

Minor concerns:

1. Could the authors provide a bit more knowledge about the R2 transelement-mediated gene insertion process in Introduction or other proper places? This can aid the readers of interest have a better understanding of the overall R2 integration working mechanism. In particular, how the second target strand is nicked? Where are the two strands nicked? Where is the template for the second strand synthesis from? And who seals the strand nicks to form an intact double-stranded duplex finally?
2. This study has determined the cryo-EM structure of R2Tg protein bound to nucleic acids, capturing nicked bottom target strand and the 3' RNA template paired to the primer target. In engineering en-R2Tg (Ref 20), multiple amino acid mutations are introduced at the protein/nucleic acid interface, based on a predicted structural model from AlphaFold. Can these predicted interactions be seen in the cryo-EM structure? From this resolved structure, importantly, which kinds of new sites might be targeted for rationally engineering more robust R2Tg transgene systems?
3. In Abstract: “We show how R2Tg binds to its DNA target and cleaves both strands in a reverse transcription-dependent manner.”. The latter statement might be toned down. To my knowledge, currently it seems that the post-TPRT process remains unclear. Meanwhile, I would suggest highlighting the compactness (small-size) and better integration efficiency of the engineered R2Tg systems compared to the previous ones.
4. In Discussion, the authors point out the potential off-target editing issue with R2Tg protein working in the absence of RNA donor. Controlling enzyme protein activity is also an important subject for the CRISPR systems (Cell 2019, 177,1067-1079.e19; J. Chem. Inf. Model. 2023, 63:3500-3509). A short statement might be presented to on the feasibility of small-molecule control over the R2Tg protein function to reduce unexpected edits.

Reviewer #3

(Remarks to the Author)

Edmonds et al. tackle one of the final genome-editing tools that is highly sought after: an all-RNA, targeted gene insertion technology. The authors begin by screening several R2 retroelement candidates to identify the ideal starting point for an all-RNA gene insertion tool, R2Tg. They leverage in vitro biochemical assays and a novel structure of R2Tg to perform rational engineering of the donor RNA; most notably, the authors dramatically reduced the size of the insertion scar through removal of the 5' UTR and a truncated 3' UTR. The authors then dramatically improved editing efficiencies through modified delivery methods involving extended incubation with LNPs. Lastly, the authors demonstrate robust on-target profiles of the optimized variants and performed editing across a variety of cell-types using their LNP delivery methods.

The authors elegantly merge structural biology, biochemical assays, and rational engineering to improve R2-mediated gene insertion efficiencies. However, it remains unclear how their work improves upon recent work published by multiple other groups. Although they cite these studies, there are no direct comparisons between the final variants in this manuscript and published variants from other works, despite several explicit claims that the described engineered system performs “more robustly.” Given that the authors work with the exact same system (R2Tg), these comparisons are critical to justify the claims made. Furthermore, the most notable improvement in editing efficiency comes from two key changes: a previously-described LNP method, and an extended transfection incubation. Thus, the authors should compare the miniMod donor described in this work with previously described R2 insertion tools, using similar LNP delivery methods. Lastly, there are several claims that the authors make that are either unsubstantiated with the current data, or would benefit from additional controls/clear references to published work (described below).

MAJOR COMMENTS

1. The authors should include direct comparisons to previously published R2Tg editors. Furthermore, the authors have the unique ability to comment on the modifications made in the previous publications (both UTR modifications and protein mutations) given their intimate structural and biochemical knowledge of R2Tg. Proper comparisons with previous works will not only strengthen the work, but also justify the claims already made by the authors.

2. Figure 1F shows smearing in all lanes in which the R2Tg and the RNA is added, but no dNTP is included. This is unexpected and would benefit from additional discussion and analysis by the authors. Additionally, this gel is used to justify the claim that top-strand is “strictly dependent on dNTPs” (line 147). The authors themselves observe that this is not entirely true, as there is clear top-strand cleavage independent of dNTP, albeit at lower rates. The language in the text should be softened to accurately reflect the data.

3. The repeated observation of CpG methylation sensitivity (paragraph starting at 131 ; lines 216, 232) is an apparent advantage of R2Tg; however, the authors include no evidence of strong CpG methylation at the native rRNA insertion site. Given R2Tg inserts at an established, well-defined locus, the authors should include data highlighting CpG methylation at these sites, leveraging publicly available data such as the UCSC genome browser.

4. The authors mention in several locations the risk of unintended DSBs due to the ability for R2Tg to cleave both strands of the target DNA, independent of functional 3' UTR RNA (lines 233-234, 325-326, 339, 322, 433, 446). However, there is only a single in vitro biochemical gel in support of this observation (Fig 1F). In vivo generation of indels through DSBs (with or without the template RNA) at rRNA loci should be shown for this claim and caution to be so heavily emphasized.

5. Notably, there is a dearth of analysis regarding two key features of R2 elements: insertion copy number per cell and inaccurate 5' insertion products (truncations, snap-backs, template jumping, etc.). Considering the authors truncated or removed UTR templates, these analyses are of the utmost importance and increasingly standard for R2 engineering publications (see Zhang et al., 2024, and Chen et al., 2024). Copy number analysis is of particular importance given the extremely high editing efficiencies observed with LNP delivery. Additionally, the authors claim that the decreased fitness may be due to TPRT-independent DSBs (line 339), while previous work (which they cited, Zhang et al., 2024) showed evidence of a reduction in copy number leading to reduced toxicity. Together, the authors explanations of toxicity appear incomplete, and significant supporting evidence should be included.

6. Given the extreme truncations to the UTR, the short homology sequence required (4nt), and the ability for R2Tg to cleave both DNA strands independent of template RNA, an obvious question is raised: are there aberrant RNA sequences incorporated to the R2Tg insertion site? The authors should be able to investigate this with their TTISS methods.

7. The off-target insertion analysis of R2Tg is impressive, but the transition from panel 5A to 5B is confusing. Panel A strongly suggests R2Tg exhibits essentially zero off-targets, while panel B samples 7,132 off-target insertion sites. This should be made more clear for the reader, as well as including a supplementary table showing the relative reads at the on-target site and all off-target sites.

8. The data supporting editing in post-mitotic myotubes cells (Extended Data Fig. 9A) is not sufficient. Luciferase assays are extremely sensitive, and a difference of ~75 luminescence units is not convincing, especially considering panel 9F shows positive signal that is 10^6 greater than the background. Furthermore, the authors often include more controls in panels, most notably ENmut. Without proper controls, this increased signal may be due to cytosolic reverse transcription and/or random insertion, both of which the authors highlighted in the text as potential risks with R2 editors. Without additional controls and supporting evidence of targeted insertion, this is not sufficient to support the claim of successful editing of post-mitotic myotubes.

9. The clear improvements in editing efficiencies are derived from a series of delivery improvements that have already been leveraged in previous work: UTP modifications, 5' capping, and LNP delivery. The authors should clearly compare their modified donor RNA to WT donor RNA with all other delivery modifications included, as well as comparing to recently published R2 insertion tools (with the same delivery modifications), to justify claims made regarding an improved and engineered R2 insertion tool.

MINOR COMMENTS

1. Lines 83 and 84 specifically reference the diversity of R2 elements with specific clades, to emphasize the diversity the authors are sampling. It would help if the authors included a tree of R2 elements, with the elements chosen highlighted, to demonstrate to the reader the diversity they are sampling. While organism silhouettes are a nice touch, the diversity of the protein itself should be clear.

2. Line 148, the authors describe dNTP requirements, but do not cite a figure. This should be clearly referenced. Furthermore, the authors say top strand cleavage was “strictly dependent on dNTPs,” yet shortly thereafter they observe 16% cleavage of the top strand in absence of dNTPs.

Version 1:

Reviewer comments:

Reviewer #1

(Remarks to the Author)

The authors have addressed all of my comments in full and I have no reservations in recommending publication.

(Remarks on code availability)

Reviewer #2

(Remarks to the Author)

The authors have addressed the majority of my previous comments. I would like to raise two additional points for their consideration:

1. In the Introduction, it is stated that the second DNA strand could be synthesized by either the R2 protein or a host polymerase. The R2 protein contains two enzymatic modules—reverse transcriptase (RT) and restriction enzyme-like enzyme (RLE). RT is known for its RNA-dependent DNA synthesis activity. The question here is that, can RT also have the ability to mediate DNA synthesis with a DNA (here, the first DNA strand) template?
2. The authors conducted additional experiments to compare the GFP gene integration efficiency of their R2Tg variants with previously reported R2 systems. The notably low efficiency observed for eg-R2Tg is quite intriguing and warrants further discussion.

(Remarks on code availability)

The code is published on GitHub and includes a README file providing instructions for installing and running the application. However, I have not yet installed or run the code myself.

Reviewer #3

(Remarks to the Author)

Edmonds et al. performed several additional experiments, analysis, and text changes that dramatically strengthened the manuscript and contribute to the field's understanding of the potential of R2 retrotransposons as a gene-insertion tool. The authors addressed a great deal of the concerns of all three reviewers, as well as incorporating text edits that accurately reflect the data presented. Notably, the authors thoroughly characterize the toxicity of various R2 constructs, demonstrating a production of indels at the rRNA loci, a loss of cell division, and only a marginal increase in gamma-H2AX levels. The revised manuscript is very thorough, and there are only a small number of remaining comments that are largely text edits.

MAJOR COMMENTS

1. The authors present a very thorough comparison of the different R2 editors in the response to reviewers. Despite their caution in interpreting the relative performances, the data are valuable to share within the final publication itself, which we encourage the authors to do in a supplementary figure. The authors can include a comment in the discussion that cautions the reader to draw conclusions of relative performance, as was included in the response.
2. The new data presented in Figure 3H is striking and warrants more discussion than the original sentence at lines 286-287. In its current form, an R2 retrotransposon has limited therapeutic utility if the functional cargo capacity is ~2.5 kb, since the coding sequence alone for most therapeutically relevant cargos are great than this size cutoff, not to mention potential regulatory elements needed. This should be transparently addressed in the discussion.

Additional questions that these data introduce are: (a) Does the introduction of the 5' UTR change the cargo tolerance, and does this suggest that 5' binding of the R2 retrotransposon improves integration efficiency? (b) Presumably, large cargos that fail to show GFP signal represent partial insertions that are due to low processivity of R2Tg, suggesting that these editing outcomes will still involve rRNA disruption and associated toxicities. This should be addressed in the discussion.

3. The authors alluded to an intriguing paradox related to R2 retrotransposon tool development, namely, whether the engineering is guided based on the wrong phenotype. The authors logically aim to improve R2 insertion via measuring % GFP positive cells 1-2 days following transfection. However, Extended Data Fig. 7 suggests that such optimizations may lead to a final R2 variant that does not support long-term viability of the host, limiting its therapeutic potential. This could be valuable to include in the discussion.

MINOR COMMENTS

1. Extended data Fig. 10: It may benefit the reader if the authors label the cargo sizes for all cargos used in this panel.

(Remarks on code availability)

Version 2:

Reviewer comments:

Reviewer #2

(Remarks to the Author)

The authors have addressed my concerns, and I have no additional comments.

(Remarks on code availability)

Reviewer #3

(Remarks to the Author)

The revised manuscript looks great and I am pleased to enthusiastically support publication. Kudos to the authors!

(Remarks on code availability)

Response to Reviews for NCOMMS-24-58727-T

Structure and biochemistry-guided engineering of an all-RNA system for DNA insertion with R2 retrotransposons

Edmonds et al.

We thank all three reviewers for their thoughtful and constructive comments. In addressing these comments, we believe the manuscript has been considerably improved and our main conclusions have been strengthened. Significant revisions include:

- **Comparison between engineered R2 systems.** To benchmark recently published R2 systems, we compared the reporter integration efficiencies of the systems described in this study (native, mini, and miniMod), Zhang et al., 2024 (ref19), and Chen et al., 2024 (ref20) using the Lipofectamine MessengerMax (MMax) and the SM102 lipid nanoparticle (LNP) delivery methods. Under these conditions, miniMod had the highest integration efficiency. The results are presented in the point-by-point response and partially presented in the new Fig. 4e,g. We also made substantial changes to the Discussion in response to the reviewer comments.
- **Characterization of the insertions.** To more extensively characterize our insertion products and to quantify their copy numbers in our LNP-delivered miniMod system, we performed Nanopore long-read sequencing and ddPCR on the genomic DNA (gDNA). The results suggest that around 53.4% of the completed integrations are full-length, which corresponds to approximately 3.3 full-length transgene copies per genome. These results, as well as other integration outcomes, are presented in the new Fig. 5d and Extended Data Figs. 8b,c.
- **Characterization of non-insertion outcomes.** To assess the rate of R2Tg-mediated DNA cleavage +/- donor RNA in cells, we performed deep sequencing on the 28S target site to detect insertions and deletions (indels). Further, we investigated R2Tg's effect on DNA damage response and cell proliferation. These results are described in the new Extended Data Fig. 7.

In addition, we made numerous other improvements to the text and figures in response to reviewer comments. We also added new Figs. 3h, 5a, Extended Data Figs. 1a, and 10a,b,g. These figures explore the effects of donor length on integration efficiency, explain the TTISS assay, illustrate R2 diversity, explore the system's ability to function in non-dividing cells, and demonstrate its ability to integrate non-reporter cargo.

Below, we respond to each reviewer comments in *italics*:

Reviewer #1 (Remarks to the Author):

In this manuscript, the authors identify an avian R2 retrotransposon (R2Tg) as active in human cells and provide a comprehensive characterization through in vitro biochemical assays and cryo-EM structure determination. The study reveals interactions of the enzyme with template RNA and target DNA, highlighting notable differences from the previously studied R2 element from *Bombyx mori* (R2Bm). Taking advantage of this structural information, the authors designed a minimal vector system with high activity in human cells. They further optimize the donor RNA structure and chemistry through base modifications, enhancing cellular stability. Combined with refined transfection protocols, this approach achieves remarkably high integration levels, reaching up to 90% under specific conditions and cell types. The manuscript presents a well-executed and clearly described body of work, with meticulously designed and controlled experiments. While R2Tg has been recently identified and repurposed for transgene integration in other studies, including all-RNA-based systems, this work offers significant new insights beyond incremental improvements (such as the cryo-EM structure of the complex, and levels of integration far above what was obtained so-far). A few important questions raised by their findings remain pending and would need to be answered before publication:

1. Off-targets and undesired events: The manuscript lacks detailed characterization of undesired events, such as indels at on-targets resulting from cleavage without insertion and subsequent NHEJ repair, truncations, junction accuracy, and potential inversions. A more comprehensive analysis of these aspects is necessary.

Response: *We agree with the reviewer on the importance of detailed characterizations of undesired edit events. To address this, we performed sequencing experiments on the gDNA of cells transfected with LNP-delivered systems:*

To characterize the rate of indel formation, we analyzed the 28S target site by deep sequencing. We found that R2Tg induced indels at the 28S locus in both the presence and absence of donor RNA (2.4 and 3.5% of the reads, respectively), and showed this result in the new Extended Data Figs. 7c,d.

To further characterize the integrations, including the undesired integrations events, we performed long-read sequencing, enriching for the 28S locus through Nanopore Cas9-targeted sequencing (Gilpatrick et al., 2020; 10.1038/s41587-020-0407-5). We found that approximately 53.4% of the integration events were full length. Truncated products resulted from what appeared to be incomplete reverse transcription and template jumping. Other events were rare, including those that contained inversions, duplications, and potential snap-backs. These results are reported in the new Fig. 5d and Extended Data Figs. 8b,c.

2. DNA damage: Does R2Tg induce detectable DNA damage, such as gamma-H2AX foci? If so, are they primarily localized to the rDNA locus? Do certain conditions (e.g., when top-strand cleavage becomes RT-independent) alter the number or location of these foci?

Response: *To determine if R2Tg's cleavage activity increased gamma-H2AX levels, we performed immunofluorescent staining on cells transfected with R2Tg, RTmut, and ENmut with and without donor RNA. R2Tg only slightly increased gamma-H2AX levels, in line with the fairly low rate of indels, with no significant difference in the presence or absence of donor RNA. These results are shown in the new Extended Data Fig. 7e.*

Related to one of reviewer #3's comments, although R2Tg did not induce a strong increase in gamma-H2AX levels, we observed that its activity slowed cell proliferation. This result is described in the new Extended Data Fig. 7f,g, and discussed in the Results in the context of decreased %GFP+ population over time.

3. Copy number: What is the average and range of transgene copy numbers under optimized conditions, given the multi-copy nature of rDNA?

Response: *To estimate the number of integrated full-length transgenes for the LNP-delivered miniMod system, we used both ddPCR and the long-read sequencing results. ddPCR results suggest that on average, R2Tg completed 3.8 ± 0.8 integrations with the expected 5' junction (mean \pm SD, per genome).*

In the long-read sequencing results, approximately 519 reads contained the sequence targeted by the ddPCR assay (contained correctly oriented binding sites for ddPCR primers and probe), while 452 reads are full-length integrations. From this ratio, we estimate approximately 3.3 copies of complete transgene copies per genome, now reported in the Results section.

4. Transgene size: What is the size limit of the donor RNA? The study only tests GFP as a transgene. How does the system perform with more relevant genes of interest, and what is the maximum size that can be accommodated while avoiding truncations?

Response: *To study the effect of donor size on integration efficiency, we systematically increased the donor size by adding "filler" sequences between the GFP expression cassette and the 3' UTR of the donor. We co-transfected the resulting constructs with R2Tg using MMax, and showed that R2Tg can integrate cargos up to ~3kb, but the efficiency decreases with increasing cargo size. This result, demonstrated with two different filler sequences, is presented in the new Fig. 3h.*

To demonstrate the ability of our system to integrate more relevant genes of interest, we integrated engineered IscB, an ω RNA-guided endonuclease (Altae-Tran et al., 2021;

10.1126/science.abj6856 and Kannan et al., under review), with miniMod donors and LNP delivery. We then introduced ω RNA and assayed for *IscB* activity by target indel sequencing. This result is reported in the new Extended Data Fig. 10g, and shows 7.2% indel rate when *IscB* donors are introduced with R2Tg (vs. 0.2% when introduced with RTmut). Together with other reporter-type donors in this figure, it demonstrates the ability of our system to integrate different cargos.

Other points:

5. The ZnF order in Fig. 2A appears to be inverted. Please verify and correct if necessary.

Response: We thank the reviewer for catching this error. Fig. 2a's ZnF is labelled according to the convention of the field, where the 3 N-terminal ZnFs are numbered 3 to 1 from N to C, and we corrected the mislabelled ZnFs in Figs. 2b,f and Extended Data Fig. 3. We also now explicitly notate this convention in Fig. 1c

6. The transfection of the TTISS donor oligo used to map off-target cleavage sites is not mentioned in the methods section. Please provide clarification.

Response: We apologize for the confusion. The existing TTISS experiment is a modified form of the assay that detects off-target insertion sites. Thus, it reads from the 3' UTR of the donor outward to the genomic DNA and does not require the transfection of additional donor oligos. We have added a new schematic to describe this experiment in Fig. 5a, and modified the description in the Results to clarify the design.

7. PDB files should be made accessible to reviewers to enable proper evaluation of the new structure and comparison with previous ones.

Response: We apologize for the missing PDB files with our first submission. We have attached the files with the resubmission.

Reviewer #2 (Remarks to the Author):

Edmonds et al. performed biochemical and structural characterization of the R2 retrotransposon system from *Taeniopygia guttata* (R2Tg). Leveraging the insights from these experiments, the authors were able to rationally engineer a compact R2Tg tool that perform more robustly than the previous native and engineered ones (as reported in Ref 19 and Ref 20, respectively) in multiple types of mammalian cells. Notably, this new system features chemical modifications as well as truncation on the donor/template RNA, while the R2-encoded “editor” protein remains intact. This work is of significance to the genome engineering field as it offers an optimized 2-RNA system for efficient, targeted gene insertion. The soundness of this research might be further enhanced if the authors could address the concerns described below.

Major concerns:

1. The authors have the 5'-UTR removed from the donor RNA to obtain a “mini donor” while keeping tens of nucleotides 5' homology. This means that there is no functional ribozyme within the 5' end of the new donor RNA. However, the recent study in Ref 20 suggests that the presence of this HDV-like ribozyme is important for R2Tg retrotransposition activity. Moreover, Ref 20 shows the necessity of at least 100 nucleotides (nt) of homology to the target locus at both the 5' and 3' ends. In the current manuscript and Ref 19, the authors demonstrate a few nts (~4) of 3' homology are sufficient for R2Tg protein-mediated integration. The apparent discrepancies between these studies on the same systems need to be clarified.

Response: *We agree there are several discrepancies between our findings and those of en-R2Tg (ref20). However, it is difficult to directly compare our system requirements due to substantial differences in our assay design.*

To our knowledge, the en-R2Tg study primarily characterized system requirements with DNA vectors expressing the R2 editor and GFP-intron reporter donor. In contrast to an all-RNA system, the plasmid expression system has different qualities that may affect integration, such as additional UTR sequences on the donor and possible differences in RNA levels, localization, post-transcriptional modifications, etc. While reporter plasmids integrated through HDR do not create GFP signal due to the intron, it may also prolong donor RNA availability over time. Thus, although both systems similarly use R2Tg, their requirements may differ.

However, we agree with the reviewer that these differences should be clarified. To address this, we made significant changes to the Discussion to explicitly point out key design differences between PRINT (ref19), en-R2Tg, and our system, focusing on the reason behind our choices and the new insights we've gained from our study. These changes are summarized below:

5' UTR and ribozyme: Our results suggest that the HDV-like ribozyme in the 5' UTR of the donor is replaceable. In contrast to R2Bm, R2Tg does not require the 5' UTR for top-strand cleavage (Fig. 1d and the new Extended Data Fig. 7c), and unlike natural R2 elements, the donor RNA does not need to process itself out of its 28S co-transcript. Instead, the ribozyme likely serves as a stable secondary structure that protects the donor RNA from degradation. Our ability to replace the R2Tg ribozyme with 5' caps (both Cap1 and modCap) (Fig. 4) and other non-Tg ribozymes (Extended Data Fig. 5d) supports this theory. Additionally, a recent publication that investigates the role of ribozymes in PRINT, which uses a *T. castaneum* R2 ribozyme, also suggests that it confers protection (Palm et al., 2024; 10.1261/rna.080031.124). Thus, we opted for the compact miniMod design with higher integration efficiency.

5' homology: Our results suggest that increasing 5' homology increases integration efficiency up to a point (Fig. 3f, up to 36 nt in the tested conditions). R2 editors likely do not require significantly longer homology, since natural R2 elements' HDV-like ribozymes process its 5' end. To our knowledge, this only leaves 0-36 nt homology on the RNA, depending on the R2 ortholog (Eickbush et al., 2013; 10.1371/journal.pone.0066441).

3' homology: In agreement with PRINT, our results suggest that R2Tg only requires 4 nt homology on the 3' end for donor integration (Fig. 3g). The low number is consistent with the resolved TPRT structure, which shows 5 nt homology (Fig. 2), and the off-target feature analysis (Fig. 5c). The use of relatively short primers for reverse transcription is also consistent with our understanding of R2Bm, which requires no homology (Luan and Eickbush, 1996; 10.1128/MCB.16.9.4726), as well as other reverse transcriptases such as human telomerase (Ghanim et al., 2021; 10.1038/s41586-021-03415-4), group II introns (Stamos et al., 2017; 10.1016/j.molcel.2017.10.024), and type 2 defense-associated reverse transcriptase (Wilkinson et al., 2024; 10.1126/science.adq3977).

Because the above points focus on design choices we made for mini and miniMod, we also further clarify, in relevant figure legends, instances where we used donors that deviate from these general designs.

2. The authors applied ψ and m1 ψ modifications to the donor RNA uridines and found that the m1 ψ modification produced higher integration efficiency than using pseudouridine (ψ) or the native uridine. It is interesting to note that an opposite trend might even be observed with different cell types, as described in Ref 20. A reasonable discussion on this point is recommended.

Response: We thank the reviewer for pointing out the interesting observation that UTP modifications may have context-dependent effects on integration efficiency. We suspect this may be due to a combination of different donor design (for example, potential effects on the ribozyme), R2 editor, and cell types – particularly in those with innate immune

responses. We now note this observation and its potential for future study in the Discussion, and point out other features of $m1\psi$ (beyond its effects on integration) that may make it a suitable modification for downstream applications.

3. It could be necessary to perform side-by-side quantitative evaluation of the new version of R2Tg with that of e.g., en-R2Tg reported in Ref 20 under same assays conditions in terms of integration efficiency and specificity. Yet, the authors can make at least a simple semi-quantitative comparison from those similar assays conducted in the studies, if present.

Response: We have edited the related Results section to more explicitly point out that our observed high integration specificity is consistent with those reported in PRINT and en-R2Tg.

To perform a side-by-side comparison with the two other described R2 systems for integration efficiency, we purchased and, where necessary, cloned the constructs described in the other two publications. We synthesized the RNAs with the modifications specified in the respective studies, and tested them in HEK293FT cells alongside our systems using both Lipofectamine MessengerMax and LNP, as suggested by reviewer #3.

“Native” is the reporter donor containing the full length R2Tg UTRs on both ends, as described in Fig. 3 of this work. en-R2Tg is described in ref20, and PRINT-Za and -Tg are described in ref19, with R2Za and R2Tg as the editor, respectively. All samples are transfected with 200 ng/well of total RNA in a 96-well format, with the optimized editor:donor ratio described in the respective studies.

Under the conditions of this experiment, miniMod showed the highest integration efficiency using both delivery modalities. PRINT's ~10% MMax efficiency closely matched what was reported in ref19 for HEK293T, and LNP transfection improved the performance of both our systems and PRINT.

While we agree with reviewers #2 and 3 that this result may strengthen our claims, we also note that our system was specifically developed and optimized under these experimental conditions, potentially giving it an advantage. Therefore, we believe it would be more informative if the side-by-side comparison were performed by an unbiased third party in a future study, ideally with more exhaustive characterization of the systems' differences in different contexts.

As such, we did not include this result in the revision, and softened our wording when discussing integration efficiency in light of the other two systems. Instead, we focus on highlighting the new insights gained from our study, as well as the compatibility of our system with current (e.g. miniMod) and future 5' RNA modification technologies as a unique advantage.

Minor concerns:

1. Could the authors provide a bit more knowledge about the R2 transelement-mediated gene insertion process in Introduction or other proper places? This can aid the readers of interest have a better understanding of the overall R2 integration working mechanism. In particular, how the second target strand is nicked? Where are the two strands nicked? Where is the template for the second strand synthesis from? And who seals the strand nicks to form an intact double-stranded duplex finally?

Response: *Absolutely. We expanded our overview of the R2-mediated integration process in the Introduction, elaborating on our current knowledge on the downstream integration process. Further, we now show R2Tg and R2Bm's top strand nicking sites in Fig. 1c.*

2. This study has determined the cryo-EM structure of R2Tg protein bound to nucleic acids, capturing nicked bottom target strand and the 3' RNA template paired to the primer target. In engineering en-R2Tg (Ref 20), multiple amino acid mutations are introduced at the protein/nucleic acid interface, based on a predicted structural model from AlphaFold. Can these predicted interactions be seen in the cryo-EM structure? From this resolved structure, importantly, which kinds of new sites might be targeted for rationally engineering more robust R2Tg transgene systems?

Response: *The en-R2Tg construct contains 5 changes compared to the WT R2Tg sequence. Two of these are insertions of chromatin modulating peptides (CMPs) that the authors hypothesized might alter chromatin accessibility to R2's favor. These insertions are in structurally disordered regions of the protein, so we cannot offer any structural insight into the effects of these insertions. The other three mutations in en-R2Tg are point mutations A793V, Q955R, and G1098R. The A793V mutation is in the RT fingers domain in the hydrophobic core of the RT fold. This mutation was performed by the authors based on multiple sequence alignments that showed a valine was more common in this position. The equivalent position in R2Bm contains a leucine without much structural alteration compared with R2Tg. We can only speculate that the valine mutation could improve hydrophobic packing to help protein folding. The Q955R and G1098R mutations were done by the authors due to the predicted proximity of these residues to the RNA-DNA duplex and upstream DNA respectively, with the aim of introducing positive charges to stabilize association. Based on our structure, we think these explanations are unlikely, as neither residue is close enough to DNA or RNA to form contacts if mutated. Instead, Q955R may form stabilizing hydrogen bonds with the Ser948 sidechain and carbonyl, and G1098R may form stabilizing hydrogen bonds with Asn1144. But these rationalizations are only speculative, and we would hesitate to nominate potential alternative sites for mutagenesis since experimental validation would be critical. We therefore did not include the above discussion in the revised manuscript.*

3. In Abstract: "We show how R2Tg binds to its DNA target and cleaves both strands in a reverse transcription-dependent manner,". The latter statement might be toned down. To my knowledge, currently it seems that the post-TPRT process remains unclear. Meanwhile, I would suggest highlighting the compactness (small-size) and better integration efficiency of the engineered R2Tg systems compared to the previous ones.

Response: *We thank the reviewer for this suggestion, and agree that the post-TPRT process remains mechanistically unclear. We modified the Abstract to tone down the statement and highlight the strengths and approaches of our system.*

4. In Discussion, the authors point out the potential off-target editing issue with R2Tg protein working in the absence of RNA donor. Controlling enzyme protein activity is also an important subject for the CRISPR systems (Cell 2019, 177,1067-1079.e19; J. Chem. Inf. Model. 2023, 63:3500-3509). A short statement might be presented to on the feasibility of small-molecule control over the R2Tg protein function to reduce unexpected edits.

Response: *We agree with the reviewer that better control over R2Tg activity will likely improve the system by reducing undesired DNA cleavage. Since the Discussion has been significantly reworked, we have added this point to the relevant new section in the Results, where we discuss R2Tg's effects on cells and, by extension, transgene population stability.*

Reviewer #3 (Remarks to the Author):

Edmonds et al. tackle one of the final genome-editing tools that is highly sought after: an all-RNA, targeted gene insertion technology. The authors begin by screening several R2 retroelement candidates to identify the ideal starting point for an all-RNA gene insertion tool, R2Tg. They leverage in vitro biochemical assays and a novel structure of R2Tg to perform rational engineering of the donor RNA; most notably, the authors dramatically reduced the size of the insertion scar through removal of the 5' UTR and a truncated 3' UTR. The authors then dramatically improved editing efficiencies through modified delivery methods involving extended incubation with LNPs. Lastly, the authors demonstrate robust on-target profiles of the optimized variants and performed editing across a variety of cell-types using their LNP delivery methods.

The authors elegantly merge structural biology, biochemical assays, and rational engineering to improve R2-mediated gene insertion efficiencies. However, it remains unclear how their work improves upon recent work published by multiple other groups. Although they cite these studies, there are no direct comparisons between the final variants in this manuscript and published variants from other works, despite several explicit claims that the described engineered system performs “more robustly.” Given that the authors work with the exact same system (R2Tg), these comparisons are critical to justify the claims made. Furthermore, the most notable improvement in editing efficiency comes from two key changes: a previously-described LNP method, and an extended transfection incubation. Thus, the authors should compare the miniMod donor described in this work with previously described R2 insertion tools, using similar LNP delivery methods. Lastly, there are several claims that the authors make that are either unsubstantiated with the current data, or would benefit from additional controls/clear references to published work (described below).

MAJOR COMMENTS

1. The authors should include direct comparisons to previously published R2Tg editors. Furthermore, the authors have the unique ability to comment on the modifications made in the previous publications (both UTR modifications and protein mutations) given their intimate structural and biochemical knowledge of R2Tg. Proper comparisons with previous works will not only strengthen the work, but also justify the claims already made by the authors.

Response:

System comparison: *We agree with this reviewer and reviewer #2 that the claims of this study could have benefited from a direct comparison with the two other R2 systems for all-RNA mediated insertion. We describe the results of this experiment below, which is also included in the response to reviewer #2:*

To perform a side-by-side comparison for integration efficiency, we purchased and, where necessary, cloned the constructs described in the other two publications. We synthesized the RNAs with the modifications specified in the respective studies, and

tested them in HEK293FT cells alongside our systems using both Lipofectamine MessengerMax and LNP, as suggested by this reviewer in a later comment.

“Native” is the reporter donor containing the full length R2Tg UTRs on both ends, as described in Fig. 3 of this work. en-R2Tg is described in ref20 (10.1016/j.cell.2024.06.020), and PRINT-Za and -Tg are described in ref19 (10.1038/s41587-024-02137-y), with R2Za and R2Tg as the editor, respectively. All samples are transfected with 200 ng/well of total RNA in a 96-well format, with the optimized editor:donor ratio described in the respective studies.

Under the conditions of this experiment, miniMod showed the highest integration efficiency using both delivery modalities. PRINT's ~10% MMax efficiency closely matched what was reported in ref19 for HEK293T, and LNP transfection improved the performance of both our systems and PRINT.

While we agree with both reviewers that this result may strengthen our claims, we also note that our system was specifically developed and optimized under these experimental conditions, potentially giving it an advantage. Therefore, we believe it would be more informative if the side-by-side comparison were performed by an unbiased third party in a future study, ideally with more exhaustive characterization of the systems' differences in different contexts.

As such, we did not include this result in the revision, and softened our wording when discussing integration efficiency in light of the other two systems. Instead, we focus on highlighting the new insights gained from our study, as well as the compatibility of our system with current (e.g. miniMod) and future 5' RNA modification technologies as a unique advantage.

In response to this suggestion, we have also modified parts of our Results and Discussion to connect the insights from our study to the design of the other two systems. These changes are summarized below:

*3' UTR modifications: Added to the Discussion, our results provide mechanistic insight into why en-R2Tg could truncate their donor's 3' UTR to 81 nt and why PRINT could use the 3' UTR of other avian orthologs (*G. fortis*). Specifically, we observed that R2Tg recognizes the pseudoknotted element in the 3' UTR for TPRT (Fig. 2e). This element is necessary and sufficient for R2Tg-mediated insertion (Fig. 3d), and its secondary structure is key to its function. Thus, en-R2Tg was able to reduce their 3' UTR to 81 nt, which retained this element. Further, the 3' UTR sequences of Tg, Za, and Gfo are highly similar, with only small differences in the 67 nt core, mostly in segments not involved in RNA folding (Extended Data Fig. 4). Thus, PRINT was able to use the 3' UTR of *G. fortis* with the editor from another ortholog.*

Protein modifications: We were only able to speculate on the effects of en-R2Tg's mutations. The en-R2Tg construct contains 5 changes compared to the WT R2Tg sequence. Two of these are insertions of chromatin modulating peptides (CMPs) that the authors hypothesized might alter chromatin accessibility to R2's favor. These insertions are in structurally disordered regions of the protein, so we cannot offer any structural insight into the effects of these insertions. The other three mutations in en-R2Tg are point mutations A793V, Q955R, and G1098R. The A793V mutation is in the RT fingers domain in the hydrophobic core of the RT fold. This mutation was performed by the authors based on multiple sequence alignments that showed a valine was more common in this position. The equivalent position in R2Bm contains a leucine without much structural alteration compared with R2Tg. We can only speculate that the valine mutation could improve hydrophobic packing to help protein folding. The Q955R and G1098R mutations were done by the authors due to the predicted proximity of these residues to the RNA-DNA duplex and upstream DNA respectively, with the aim of introducing positive charges to stabilize association. Based on our structure, we think these explanations are unlikely, as neither residue is close enough to DNA or RNA to form contacts if mutated. Instead, Q955R may form stabilizing hydrogen bonds with the Ser948 sidechain and carbonyl, and G1098R may form stabilizing hydrogen bonds with Asn1144. But these rationalizations are only speculative. As such, we did not include its discussion in the revised manuscript.

On the other hand, we provide additional insight into why PRINT's "EN-tuned" (ENT) editors maintained a stable transgene population over time. Specifically, while characterizing R2Tg's effect on cell fitness (in response to a later reviewer comment), we observed that the transgene-positive population likely decreased with time due to their slower cell division rates. Our results further suggest that R2Tg-mediated DNA cleavage can contribute to this decreased growth rate. This partly explains why ENT, a mutant with lower endonuclease activity, gave rise to a relatively more stable transgene-

positive population than their WT counterpart, albeit at the cost of integration efficiency. This is added to the Results when discussing the new Extended Data Fig. 7.

2. Figure 1F shows smearing in all lanes in which the R2Tg and the RNA is added, but no dNTP is included. This is unexpected and would benefit from additional discussion and analysis by the authors. Additionally, this gel is used to justify the claim that top-strand is “strictly dependent on dNTPs” (line 147). The authors themselves observe that this is not entirely true, as there is clear top-strand cleavage independent of dNTP, albeit at lower rates. The language in the text should be softened to accurately reflect the data.

Response: *We thank the reviewer for pointing this out. The slight smearing in TPRT gels can be reduced by phenol extracting the reactions before gel loading, so likely derives from denatured R2Tg protein binding to nucleic acids at a low level. While phenol extracting all lanes before loading could be desirable, the variable DNA recovery sometimes interfered with quantitative assessment of TPRT efficiency, so we elected to load non-deproteinized reactions onto the gels for easier reproducibility. We don't think that the background smearing interferes with assessment of the reaction outcomes, and have added an explanation to the Fig. 1f legend.*

For the second point, we agree there is a low level of top strand cleavage in the absence of dNTPs. We deleted the word “strictly” to more accurately reflect the results, but don't think further wording change is necessary given that the precise efficiency is quantified in the text.

3. The repeated observation of CpG methylation sensitivity (paragraph starting at 131; lines 216, 232) is an apparent advantage of R2Tg; however, the authors include no evidence of strong CpG methylation at the native rRNA insertion site. Given R2Tg inserts at an established, well-defined locus, the authors should include data highlighting CpG methylation at these sites, leveraging publicly available data such as the UCSC genome browser.

Response: *We agree with the reviewer that evidence for CpG methylation at R2 insertion sites could strengthen our discussion on the benefits of R2Tg vs. R2Bm. To our understanding, rDNA loci are poorly represented in genome assemblies prior to T2T due to its highly repetitive nature, and conventional pipelines often omit or inadequately map the 28S regions. Consequently, publicly available datasets sometimes lack robust annotations for the 28S locus. To address this, we referenced two recent studies that specifically re-analyzed rDNA methylation datasets, illustrating high variability in CpG methylation at the 28S locus: Zhang et al., 2022 (10.1093/bib/bbac278) reanalyzed human bisulfite sequencing data with a novel mapping strategy designed for rDNA, showing substantial tissue-to-tissue variations in 28S methylation. Wang and Lemos, 2019 (10.1101/gr.241745.118) examined rDNA methylation in multiple species, showing that 28S methylation levels can vary with age.*

In light of these studies, we softened our language. Instead of emphasizing R2Tg's insensitivity to CpG methylation as a strict advantage, we present it as a feature that increases its flexibility in mammalian cells, which may have context-dependent variance in CpG methylation patterns.

4. The authors mention in several locations the risk of unintended DSBs due to the ability for R2Tg to cleave both strands of the target DNA, independent of functional 3' UTR RNA (lines 233-234, 325-326, 339, 322, 433, 446). However, there is only a single in vitro biochemical gel in support of this observation (Fig 1F). In vivo generation of indels through DSBs (with or without the template RNA) at rRNA loci should be shown for this claim and caution to be so heavily emphasized.

Response: *We agree with the reviewer that this claim should be supported by in vivo data. In response to this and reviewer #1's comment, we assayed for R2Tg's ability to cut and generate indels at its 28S target through deep-sequencing. This result, which shows elevated indel generation by the wildtype protein both with and without donor RNA (2.4 and 3.5%, respectively), is presented in the new Extended Data Fig. 7c.*

5. Notably, there is a dearth of analysis regarding two key features of R2 elements: insertion copy number per cell and inaccurate 5' insertion products (truncations, snap-backs, template jumping, etc.). Considering the authors truncated or removed UTR templates, these analyses are of the utmost importance and increasingly standard for R2 engineering publications (see Zhang et al., 2024, and Chen et al., 2024). Copy number analysis is of particular importance given the extremely high editing efficiencies observed with LNP delivery. Additionally, the authors claim that the decreased fitness may be due to TPRT-independent DSBs (line 339), while previous work (which they cited, Zhang et al., 2024) showed evidence of a reduction in copy number leading to reduced toxicity. Together, the authors explanations of toxicity appear incomplete, and significant supporting evidence should be included.

Response:

Copy number and insertion products: *We agree with this reviewer and reviewer #1 that we needed analysis on the insertion copy number, 5' junction, as well as insertion accuracy in general. To address this, we performed long read sequencing to analyze the insertion products, enriching for the 28S locus through Nanopore Cas9-targeted sequencing (Gilpatrick et al., 2020; 10.1038/s41587-020-0407-5). We found that approximately 53.4% of the integration events were full length. Truncated products resulted from what appeared to be incomplete reverse transcription and template jumping. Other events were rare, including those that contained inversions, duplications, and potential snap-backs. These results are reported in the new Fig. 5d and Extended Data Figs. 8b,c.*

To estimate the number of integrated full-length transgenes for the LNP-delivered miniMod system, we used both ddPCR and the long-read sequencing results. ddPCR results suggest that on average, R2Tg completed 3.8 ± 0.8 integrations with the expected 5' junction (mean \pm SD, per genome). In the long-read sequencing results, approximately 519 reads contained the sequence targeted by the ddPCR assay (contained correctly oriented binding sites for ddPCR primers and probe), while 452 reads are full-length integrations. From this ratio, we estimate approximately 3.3 copies of complete transgene copies per genome, now reported in the Results section.

Cell fitness: We suspect that the decrease in %GFP+ cells is at least in part due to both R2Tg-mediated insertions (consistent with Zhang et al., 2024) and cleavage. We agree with the reviewer that we did not present sufficient data to support this theory. To address this, we made the following changes:

To show the relationship between changes in %GFP+ cells and integration, we added to Extended Data Fig. 6c,d to follow the median fluorescence intensity of the GFP+ population in the longitudinal study previously described in the figure (using MFI as a proxy for insertion number, described in Zhang et al., 2024). We observed a notable drop in GFP intensity in the conditions that also displayed high R2Tg activity (e.g. those with higher editor amounts).

To further explain the decrease in %GFP+ cells, we show that it is partly due to slower cell division of the GFP+ cells by using dye dilution assays (CellTrace). This result is shown in the new Extended Data Fig. 7b.

In addition to insertion, R2Tg-mediated DNA cleavage may also compromise cell growth. To investigate this possibility, we followed the proliferation of cells transfected with R2Tg, RTmut, and ENmut with and without donor RNA using the same CellTrace assay. We demonstrated that regardless of the presence of donor RNA, R2Tg reduced cell division compared to cells transfected with ENmut. This result is shown in the new Extended Data Fig. 7f,g. Interestingly, this also provides new insight into why the PRINT-ENT R2, a mutant editor that traded editing efficiency for transgene stability, harbors a mutation affecting the endonuclease domain.

Together, these results suggest that the observed decrease in %GFP+ cells may be due to a culmination of R2-mediated insertion, DNA cleavage, indel formation, and other disruptions to the 28S. Future studies may further unravel the relative contributions of each perturbation. We have adjusted the writing in the Results to reflect this point.

6. Given the extreme truncations to the UTR, the short homology sequence required (4nt), and the ability for R2Tg to cleave both DNA strands independent of template RNA, an obvious question is raised: are there aberrant RNA sequences incorporated to the R2Tg insertion site? The authors should be able to investigate this with their TTISS methods.

Response: From the indel length analysis, we observed that the vast majority of small changes to the 28S locus are deletions rather than insertions (Extended Data Fig. 7d). We did not observe any large (≥ 50 bp), non-donor insertions from our long-read sequencing analysis.

7. The off-target insertion analysis of R2Tg is impressive, but the transition from panel 5A to 5B is confusing. Panel A strongly suggests R2Tg exhibits essentially zero off-targets, while panel B samples 7,132 off-target insertion sites. This should be made more clear for the reader, as well as including a supplementary table showing the relative reads at the on-target site and all off-target sites.

Response: Our previous presentation of the data was confusing. To address this, we regenerated panels B and C (previously A and B, respectively) to improve their clarity. Data in the two panels are now analyzed by the same pipeline, use T2T for genome mapping (previously, GRCh38), and are graphically connected to indicate that the off-target sites are derived from the MMax data. We removed the old Extended Data Figure panels that showed redundant information to the new Fig. 5b, and added Supplementary Table 3 to show all the detected off-target sites.

8. The data supporting editing in post-mitotic myotubes cells (Extended Data Fig. 9A) is not sufficient. Luciferase assays are extremely sensitive, and a difference of ~ 75 luminescence units is not convincing, especially considering panel 9F shows positive signal that is 10^6 greater than the background. Furthermore, the authors often include more controls in panels, most notably ENmut. Without proper controls, this increased signal may be due to cytosolic reverse transcription and/or random insertion, both of which the authors highlighted in the text as potential risks with R2 editors. Without additional controls and supporting evidence of targeted insertion, this is not sufficient to support the claim of successful editing of post-mitotic myotubes.

Response: To address this, we repeated the experiment with LNP delivered miniMod nanoluciferase donor (as in panel F), adding the “ENmut” and “donor only” negative controls. The result yields a positive signal around 10^5 above background and is reported in the new Extended Data Fig. 10a.

Related, we updated the aphidicolin experiment (previously Extended Data Fig. 9b) with the miniMod donor (previously an older system) and triplicates (previously duplicates). The new results do not change the conclusion of the experiment, and are reported in Extended Data Fig. 10b.

9. The clear improvements in editing efficiencies are derived from a series of delivery improvements that have already been leveraged in previous work: UTP modifications, 5' capping, and LNP delivery. The authors should clearly compare their modified donor RNA to WT donor RNA with all other delivery modifications included, as well as comparing to recently published R2 insertion tools (with the same delivery modifications), to justify claims made regarding an improved and engineered R2 insertion tool.

Response: *We agree with the reviewer that the manuscript would benefit from a more direct comparison between the progressive improvements made to the system. To do so, we show the side-by-side comparison between “native”, “mini”, and “miniMod” donors for both MMax and LNP deliveries (subset of the previously discussed experiment that compared the published R2 systems). These results are presented in the new Figs. 4e,g, and demonstrate that miniMod improved upon mini, which improved upon native, in both delivery modalities. Please see above regarding the comparison to other published R2 insertion tools.*

MINOR COMMENTS

1. Lines 83 and 84 specifically reference the diversity of R2 elements with specific clades, to emphasize the diversity the authors are sampling. It would help if the authors included a tree of R2 elements, with the elements chosen highlighted, to demonstrate to the reader the diversity they are sampling. While organism silhouettes are a nice touch, the diversity of the protein itself should be clear.

Response: *We appreciate the suggestion to add a tree to illustrate the diversity of the R2 elements, and have incorporated this in the new Extended Data Fig. 1a.*

2. Line 148, the authors describe dNTP requirements, but do not cite a figure. This should be clearly referenced. Furthermore, the authors say top strand cleavage was “strictly dependent on dNTPs,” yet shortly thereafter they observe 16% cleavage of the top strand in absence of dNTPs.

Response: *We thank the reviewer for pointing this out. We have added a reference to Fig. 1f, and have deleted the word “strictly”.*

Response to Reviews for NCOMMS-24-58727A

Structure and biochemistry-guided engineering of an all-RNA system for DNA insertion with R2 retrotransposons

Edmonds et al.

We thank all three reviewers for their follow-up comments. We updated the manuscript to address the comments and improve the clarity of data representation (Fig. 3h). We also made minor corrections to the GitHub deposition to accurately reflect the latest version used to generate the final figures.

We address the suggestions below in *italics*:

Reviewer #1 (Remarks to the Author):

The authors have addressed all of my comments in full and I have no reservations in recommending publication.

Response: *We are grateful for all the reviewers' comments, which have significantly improved the manuscript.*

Reviewer #2 (Remarks to the Author):

The authors have addressed the majority of my previous comments. I would like to raise two additional points for their consideration:

Response: *We appreciate the reviewer's previous comments, and address the additional points below.*

1. In the Introduction, it is stated that the second DNA strand could be synthesized by either the R2 protein or a host polymerase. The R2 protein contains two enzymatic modules—reverse transcriptase (RT) and restriction enzyme-like enzyme (RLE). RT is known for its RNA-dependent DNA synthesis activity. The question here is that, can RT also have the ability to mediate DNA synthesis with a DNA (here, the first DNA strand) template?

Response: *We thank the reviewer for highlighting this interesting point. Studies on R2Bm suggest that the R2-RT can indeed act as a DNA-dependent DNA polymerase (Kurzynska-Kokorniak et al., 2007; 10.1016/j.jmb.2007.09.047 and Dangerfield et al., 2024; 10.1101/2024.08.28.610141). We have revised the relevant section of the manuscript and added the corresponding references in the Introduction to reflect this information.*

2. The authors conducted additional experiments to compare the GFP gene integration efficiency of their R2Tg variants with previously reported R2 systems. The notably low efficiency observed for eg-R2Tg is quite intriguing and warrants further discussion.

Response: *We do not know the reason behind en-R2Tg's observed efficiency. However, we can speculate that designs unique to en-R2Tg (but are otherwise similar between PRINT and mini/miniMod) may contribute to its lower efficiency under this experimental condition (e.g., poly(A) tail length, design of the R2Tg expression construct and cargo expression construct, homology length, and variations in the R2Tg protein). Many of these aspects are discussed throughout the text.*

Reviewer #2 (Remarks on code availability):

The code is published on GitHub and includes a README file providing instructions for installing and running the application. However, I have not yet installed or run the code myself.

Reviewer #3 (Remarks to the Author):

Edmonds et al. performed several additional experiments, analysis, and text changes that dramatically strengthened the manuscript and contribute to the field's understanding of the potential of R2 retrotransposons as a gene-insertion tool. The authors addressed a great deal of the concerns of all three reviewers, as well as incorporating text edits that accurately reflect the data presented. Notably, the authors thoroughly characterize the toxicity of various R2 constructs, demonstrating a production of indels at the rRNA loci, a loss of cell division, and only a marginal increase in gamma-H2AX levels. The revised manuscript is very thorough, and there are only a small number of remaining comments that are largely text edits.

Response: *We appreciate the positive response, and aim to address the remaining concerns through text edits and an additional experiment, described below.*

MAJOR COMMENTS

1. The authors present a very thorough comparison of the different R2 editors in the response to reviewers. Despite their caution in interpreting the relative performances, the data are valuable to share within the final publication itself, which we encourage the authors to do in a supplementary figure. The authors can include a comment in the discussion that cautions the reader to draw conclusions of relative performance, as was included in the response.

Response: *We thank the reviewer for the suggestion. After extensive deliberation, we remain concerned that including this data, even with the appropriate emphasis on the caveats, could lead to overinterpretation or unintended bias. As such, we elected to omit*

it from the final paper (although it will be available via the reviewer comments and responses).

2. The new data presented in Figure 3H is striking and warrants more discussion than the original sentence at lines 286-287. In its current form, an R2 retrotransposon has limited therapeutic utility if the functional cargo capacity is ~2.5 kb, since the coding sequence alone for most therapeutically relevant cargos are great than this size cutoff, not to mention potential regulatory elements needed. This should be transparently addressed in the discussion.

Response: *We agree that cargo capacity is an important consideration. We have added a new paragraph to the Discussion to emphasize this point in light of the results presented in Fig. 3h.*

Additional questions that these data introduce are: (a) Does the introduction of the 5' UTR change the cargo tolerance, and does this suggest that 5' binding of the R2 retrotransposon improves integration efficiency? (b) Presumably, large cargos that fail to show GFP signal represent partial insertions that are due to low processivity of R2Tg, suggesting that these editing outcomes will still involve rRNA disruption and associated toxicities. This should be addressed in the discussion.

Response:

(a) To determine whether the 5' UTR affects cargo tolerance, we re-introduced the full length 5' UTR to a subset of mini donor constructs used in Fig. 3h and tested its effects on integration. The results, shown in the new Extended Data Fig. 4i, suggest that the 5' UTR does not improve the cargo tolerance of the system.

(b) We agree it is very likely that partial insertions substantially lower the integration efficiency of large cargoes. We now address this in the Discussion, both in the context of cargo-size limitation and cell proliferation/viability.

3. The authors alluded to an intriguing paradox related to R2 retrotransposon tool development, namely, whether the engineering is guided based on the wrong phenotype. The authors logically aim to improve R2 insertion via measuring % GFP positive cells 1-2 days following transfection. However, Extended Data Fig. 7 suggests that such optimizations may lead to a final R2 variant that does not support long-term viability of the host, limiting its therapeutic potential. This could be valuable to include in the discussion.

Response: *This is an important consideration, especially for studies that aim to improve the editor activity. We have added a new paragraph in the Discussion to highlight this point, and restructured an existing paragraph to suggest retargeting as an additional, potential solution to this problem.*

MINOR COMMENTS

1. Extended data Fig. 10: It may benefit the reader if the authors label the cargo sizes for all cargos used in this panel.

Response: *We agree, and have added the cargo sizes of each donor to the figure legends of Extended Data Fig. 10.*